# Observations of the vertical distributions of summertime atmospheric pollutants in Nam Co: OH production and source analysis

Chengzhi Xing[1], Cheng Liu[1,2,3,4,*], Chunxiang Ye[5,*], Jingkai Xue[6], Hongyu Wu[6], Xiangguang Ji[7], Jinping Ou[8], and Qihou Hu[1]

[1] Key Lab of Environmental Optics & Technology, Anhui Institute of Optics and Fine Mechanics, Hefei Institutes of Physical Science, Chinese Academy of Sciences, Hefei, 230031, China

[2] Department of Precision Machinery and Precision Instrumentation, University of Science and Technology of China, Hefei, 230026, China

[3] Center for Excellence in Regional Atmospheric Environment, Institute of Urban Environment, Chinese Academy of Sciences, Xiamen, 361021, China

[4] Key Laboratory of Precision Scientific Instrumentation of Anhui Higher Education Institutes, University of Science and Technology of China, Hefei, 230026, China

[5] College of Environmental Sciences and Engineering, Peking University, 100871 Beijing

[6] School of Environmental Science and Optoelectronic Technology, University of Science and Technology of China, Hefei, 230026, China

[7] Institute of Physical Science and Information Technology, Anhui University, Hefei, 230601, China

[8] The Department of Health Promotion and Behavioral Sciences, School of Public Health, Anhui Medical University, Hefei, 230032, China

*Corresponding author. E-mail: chliu81@ustc.edu.cn; c.ye@pku.edu.cn

**Abstract**
The Tibetan Plateau (TP) plays a key role in regional environment and global climate change, however,
the lack of vertical observation of atmospheric species, such as HONO and $O_3$, hinders a deeper
understanding of the atmospheric chemistry and atmospheric oxidation capacity (AOC) on the TP. In
this study, we conducted multi-axis differential optical absorption spectroscopy (MAX-DOAS)
measurements at Nam Co, the central TP, to observe the vertical profiles of aerosol, water vapor ($H_2O$),
$NO_2$, HONO and $O_3$ from May to July 2019. In addition to $NO_2$ mainly exhibiting a Gaussian shape
with the maximum value appearing at 300-400 m, other four species all showed an exponential shape
and decreased with the increase of height. The maximum values of monthly averaged aerosol (0.17
$km^{-1}$) and $O_3$ (66.71 ppb) occurred on May, $H_2O$ ($3.68 \times 10^{17}$ molec $cm^{-3}$) and HONO (0.13 ppb)
appeared on July, while $NO_2$ (0.39 ppb) occurred on June at 200-400 m layer. $H_2O$, HONO and $O_3$ all
exhibited a multi-peak pattern, and aerosol appeared a bi-peak pattern for their averaged diurnal
variations. The averaged vertical profiles of OH production rates from $O_3$ and HONO all exhibited an
exponential shape decreasing with the increase of height with maximum values of 2.61 ppb/h and 0.49
ppb/h at the bottom layer, respectively. The total OH production rate contributed by HONO and $O_3$ on
the TP was obviously larger than that in low-altitude areas. In addition, source analysis for HONO and
$O_3$ at different height layers were conducted. The heterogeneous reaction of $NO_2$ on wet surfaces was a
significant source of HONO. The maximum values of HONO/$NO_2$ appeared around $H_2O$ being $1.0 \times$
$10^{17}$ molec $cm^{-3}$ and aerosol being lager 0.15 $km^{-1}$ under 1.0 km, and the maximum values usually
accompanied with $H_2O$ being $1.0-2.0 \times 10^{17}$ molec $cm^{-3}$ and aerosol being lager 0.02 $km^{-1}$ at 1.0-2.0 km.
$O_3$ was potentially sourced from south Asian subcontinent and Himalayas through long-range transport.
Our results enrich the new understanding of vertical distribution of atmospheric components and
explained the strong AOC on the TP.
**1 Introduction**
The TP spans 2.5 million square kilometers with an average altitude of over 4000 m. Therefore, the TP
is called the "Third Pole" of the earth (Ma et al., 2020; Kang et al., 2022). It is the home to tens of
thousands of glaciers and nourishes more than 10 of Asia's rivers, thus it also acts the role of "Water
Tower of Asia" (Qu et al., 2019; Ma et al., 2022). Due to its special topography, the TP is the heat
source of atmosphere due the strong solar radiation, which as the driven force to profoundly affect the
regional atmospheric circulation, global weather conditions and climate change (Yanai et al., 1992;
Boos et al., 2010; Chen et al., 2015; Liu et al., 2022; Zhou et al., 2022). Monsoon rainfall in Asia, flood
over the Yangtze River valley, and El Niño in the Pacific Ocean are strongly associated with the TP
(Hsu et al., 2003; Li et al., 2016; Lei et al., 2019). In addition, the cyclone circulations caused by the
TP heat source also can inhibit the diffusion of atmospheric pollutants in the areas around the TP, such
as the Sichuan Basin, causing regional pollution (Zhang et al., 2019). Therefore, observations of the
atmospheric species on the TP are essential to enhance the in-depth understanding of its atmospheric
physicochemical processes.
However, deciphering the atmospheric environment of the TP is highly challenging and dangerous, due
to its complex topography and harsh environment (Barnett et al., 2005; Bolch et al., 2012; Cong et al.,
2015; Kang et al., 2016). In order to unveil the feature of atmospheric composition over the TP and
their corresponding climate feedback, a large number of field observation stations have been
established, and a series of field campaigns have continued to be carried out recently, especially after
the performance of "the Second Tibetan Plateau Scientific Expedition and Research Program" (Che and
Zhao 2021; Wang et al., 2021; Ran et al., 2022). The China National Environmental Monitoring Center
(CNEMC) has established an in-situ monitoring network with more than 12 stations over the TP, such
as Lhasa, Shigatse, Shannan, Nyingchi, Nagqu, Ngari, Qamdo, Diqing, Aba, Guoluo, Xining, and
Haixi, to continuously monitor the surface concentrations of six atmospheric components (i.e. $PM_{10}$,
$PM_{2.5}$, $NO_2$, $SO_2$, $O_3$ and CO) since 2013 (Gao et al., 2020; Li et al., 2020; Sun et al., 2021). The
Institute of Tibetan Plateau Research, Chinese Academy of Sciences, has also established six long-term
field observation stations to measure meteorological parameters and small amounts of atmospheric
composition (i.e. black carbon, aerosol optical density (AOD)) (Ma et al., 2020). In addition, scientists
are relying on advancements in satellite remote sensing technology, such as the tropospheric
monitoring instrument (TROPOMI), the ozone monitoring instrument (OMI), the moderate-resolution
imaging spectroradiometer (MODIS) and the cloud-aerosol lidar and infrared pathfinder satellite
observation (CALIPSO), to monitor the spatial and temporal evolutions of atmospheric composition on
the TP (Zhu et al., 2019; Li et al., 2020; Rawat and Naja 2022). Their advantage is to obtain the column
densities of pollutants in a large-scale space of the TP. Although CALIPSO could detect aerosol
vertical profiles, the spatiotemporal resolution (i.e. ~5.0 km horizontal resolution, 0.06 km vertical
resolution and ~16 d temporal resolution) is limited and the data uncertainty in the planetary boundary
layer (PBL) is large due to the low signal-to-noise ratio (Huang et al., 2007). However, several studies
also revealed that the formation, aging and transport processes of atmospheric composition on the TP
occurs not only near the ground surface but also at high altitudes (Xu et al., 2020; Xu et al., 2022). The
high PBL on the TP caused by its strong solar radiation and undulating terrain promotes the
atmospheric exchange between the bottom troposphere and stratosphere (Yang et al., 2003; Seidel et al.,
2010). Therefore, the lack of vertical profiles of hinders the understanding of the evolution of trace
gases and their environmental and climate effects over the TP. In recent years, balloon and lidar
vertical measurements on the TP are occasionally carried out (Fang et al., 2019; Zhang et al., 2020;
Dong et al., 2022), but their limited detection species (i.e. aerosol and $O_3$) and high cost are obstacles
that limit long-term continuous observation and the conduction of more in-depth scientific research.
MAX-DOAS has the technical advantage of low-cost continuous observation of multiple atmospheric
components (i.e. aerosol, $O_3$ and their precursors) (Wang et al., 2018; Ma et al., 2020; Cheng et al.,
2021; Xing et al., 2021; Li et al., 2022; Cheng et al., 2023a, 2023b). Combining these data with better
scientific models can reduce the modeling bias and promote to better understand the physical, chemical
and dynamical processes.
The strong convergent airflow formed under the combined action of monsoon, subtropical anticyclone
and the airflow of subtropical westerlies could promote the accumulation of $O_3$ on the TP in summer
(Ye and Gao 1997). Therefore, several studies have revealed the high $O_3$ concentration on the TP (Li et
al., 2022; Yang et al., 2022; Yu et al., 2022). The strong solar radiation, high $O_3$ concentration and
relatively high humidity on the TP provide great potential for high OH production. Lin et al. (2008) and
Ye (2019) also confirmed that the high OH over the TP is mainly related to the reaction between $O(^1D)$
and $H_2O$. The $O(^1D)$ is produced from the photolysis of $O_3$ by UV radiation. Therefore, a hypothesis of
"strong AOC over the TP" was put forward. Previous studies pointed out that HONO also play an
important role in AOC at low-altitude areas, and its contribution to OH can reach 40-60%, and even
more than 80% in the early morning (Michoud et al., 2012; Ryan et al., 2018; Xue et al., 2020).
However, few HONO studies on the TP have been reported. Our previous study operated at the
Qomolangma Atmospheric and Environmental Observation and Research Station, Chinese Academy of
Sciences (QOMS-CAS) revealed that the HONO mainly distributed in the lower PBL and peaked in
summer with 1.11 ppb, which is comparable to the average level of HONO in other low-altitude areas
(Luo et al., 2010; Xing et al., 2021a, 2021b; Yang et al., 2021). It indicates that it is also necessary to
study the contribution of HONO to AOC on the TP. Furthermore, understanding the vertical
distribution of OH is of great significance for learning about the atmospheric chemical processes and
the evolution of atmospheric components on the TP (Zhou et al., 2015). Identifying the sources of $O_3$
and HONO is the basis for studying the AOC on the TP. The limited researches concluded that the
atmospheric HONO on the TP is mainly sourced from the emissions of vehicles, biomass burning and
soil, except for the $NO_2$ heterogeneous reaction on aerosol surfaces (Xing et al., 2021). The lower
tropospheric $O_3$ on the TP is mainly dominated by local photochemical reactions, regional horizontal
transport, vertical mixing and the intrusion from stratosphere (Yin et al., 2017; Xu et al. 2018).
In this study, we firstly analyzed the temporal and vertical characteristics of several atmospheric
components (i.e. aerosol, $H_2O$, $NO_2$, HONO and $O_3$) based on MAX-DOAS observations in Nam Co.
Afterwards, the contributions of $O_3$ and HONO to OH in the vertical space were discussed through the
tropospheric ultraviolet and visible (TUV) radiative transfer model and MAX-DOAS measurements.
Finally, the potential sources of $O_3$ and HONO at different altitudes were analyzed based on the
MAX-DOAS retrievals.

## 2 Method and methodology

### 2.1 Site

The Nam Co Monitoring and Research Station for Multisphere Interactions, CAS (NAMORS)
(30.774ºN, 90.988ºE; 4730 m a.s.l.) is located at the southeast banks of Nam Co lake and the foothills
of the northern Mt. Nyainqêntanglha (Fig. 1). The station land is covered by alpine meadows with soil
type of sandy silt loam. The southwest monsoon can carry abundant moisture from Indian Ocean to this
station in summer to increase humidity and precipitation there. Moreover, due to the summertime huge
evaporation from Nam Co lake, the atmospheric $H_2O$ around CAS (NAMORS) is more abundant than
in other areas of the TP, resulting in lush grass vegetation and making the area around this station an
important summertime pasture. In addition, there are not large industries and cities within 100 km of
the CAS (NAMORS). The closest town to CAS (NAMORS) is Dangxiong county which is about 60
km away from this station and lower about 500 m than this station. Only a small number of vehicles
pass through this area during summer tourism season. Therefore, no obvious anthropogenic sources of
air pollutants exist near this station. Averaged spatial distributions of AOD, $O_3$, $NO_2$ and HCHO
monitored by satellite from May to July 2019 are shown in Figure S1. Elevated AOD, $NO_2$, and $O_3$ are
mainly distributed in South Asian subcontinent (e.g. India and Nepal), the southern foothills of the
Himalayas, which is located in the upwind direction of the southwest monsoon potentially affecting the
atmospheric composition over CAS (NAMORS).

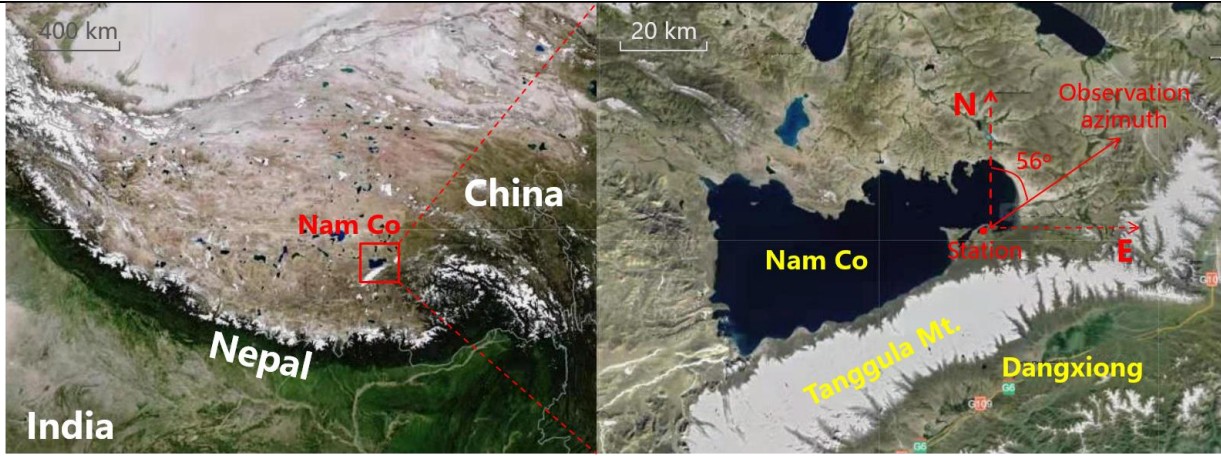


Figure 1. Geographical location of CAS (NAMORS) on the Tibet plateau.
**2.2 Measurements**
2.2.1 Instrument setup and spectral analysis
The MAX-DOAS instrument installed at CAS (NAMORS) was operated from 01 May to 09 July 2019.
It consists of three major parts: telescope unit, spectrometer unit and control unit. The detailed
description of this instrument can be found in Xing et al. (2021). In this study, the elevation angle
sequence was set to 1, 2, 3, 4, 5, 6, 8, 10,15, 30, and 90$^\circ$ with an exposure time of 60 s to each
individual spectrum. The azimuth angle was set to 56$^\circ$ pointing to Nagqu direction. Moreover, only
spectra collected under solar zenith angle (SZA) less than 75$^\circ$ was used for spectral analysis to avoid
the strong stratospheric absorption.
The differential slant column densities (DSCDs) of $O_4$, $H_2O$, $NO_2$, HONO and $O_3$ were retrieved using
QDOAS software (http://uvvis.aeronomie.be/software/QDOAS/) developed by Royal Belgian Institute
for Space Aeronomy (BIRA-IASB). The zenith spectrum measured at every sequence were selected as
scan Frauenhofer reference spectrum. The retrieval configurations of $O_4$, $H_2O$, $NO_2$, HONO and $O_3$
followed Xing et al. (2017), Lin et al. (2020), Xing et al. (2021), Wang et al. (2020) and Wang et al.
(2018), respectively. The detailed DOAS fit settings of above five species were listed in Table 1.
Corrected $I_0$ (Aliwell et al., 2002) was used in this study. Fig. 2 shows a typical DOAS retrieval
example for above five species. DOAS fit results with root mean square (RMS) values larger than $5\times$
$10^{-4}$, $5\times10^{-4}$, $5\times10^{-4}$, $1\times10^{-3}$, and $6\times10^{-4}$ for $O_4$, $H_2O$, $NO_2$, HONO, and $O_3$, respectively, were
filtered out. In addition, we calculated color index (CI) to remove cloud effect (Wagner et al., 2016).
The data filter criteria according to CI followed by Ryan et al. (2018) and Xing et al. (2020).
Afterwards, the quantified DSCDs of $O_4$, $H_2O$, $NO_2$, HONO, and $O_3$ remained 91.33%, 91.97%,
92.16%, 86.42% and 81.09%, respectively.
2.2.2 Vertical profile retrieval
The vertical profiles of aerosol and trace gases (i.e. $H_2O$, $NO_2$, HONO and $O_3$) were retrieved using
algorithm based on optimal estimation method (OEM). A linearized pseudo-spherical vector discrete
ordinate radiative transfer model VLIDORT was used as forward model and a Gauss-Newton (GN)
scheme was used as the inversion strategy (Wedderburn et al., 1974). The detailed description of this
algorithm can be found in Liu et al. (2021), Xing et al. (2021) and Wang et al. (2018). The detailed
retrieval processes were depicted in Sect. S1 of the supplement. In this study, the initial a priori profile
shape of above five species was set to exponential decreasing shape, and the AOD and vertical column
densities (VCDs) simulated by weather research and forecasting model coupled chemistry (WRF-Chem)
were also used as initial input a priori information to constrain the retrieval process. For the $O_3$ profile
retrieval, the stratospheric $O_3$ profile was deducted using TROPOMI $O_3$ profile (Zhao et al., 2021). We
set 20 vertical layers from 0.0 to 4.0 km with a vertical resolution of 0.2 km. The correlation height was
set to 1.0 km. Moreover, the surface albedo, single scattering albedo and asymmetry parameter were set
to fixed constant of 0.08, 0.85 and 0.65, respectively (Irie et al., 2008). The retrieved vertical profiles

were removed under the condition of degree of freedom (DOF) and relative error less than 1.0 and 100%, respectively.

Table 1. Detailed DOAS retrieval settings for $O_4$, $H_2O$, $NO_2$, HONO and $O_3$.

| Parameter | Data source | Fitting intervals (nm) | | | | |
|---|---|---|---|---|---|---|
| | | $O_4$ | $H_2O$ | $NO_2$ | HONO | $O_3$ |
| Wavelength range | | 338-370 | 433-455 | 338-370 | 340-373 | 320-340 |
| $NO_2$ | 298K, $I_0$-corrected, Vandaele et al. (1998) | √ | √ | √ | √ | √ |
| $NO_2$ | 220K, $I_0$-corrected, Vandaele et al. (1998) | √ | √ | √ | √ | × |
| $O_3$ | 223K, $I_0$-corrected, Serdyuchenko et al. (2014) | √ | √ | √ | √ | √ |
| $O_3$ | 243K, $I_0$-corrected, Serdyuchenko et al. (2014) | √ | × | √ | √ | × |
| $O_3$ | 293K, $I_0$-corrected, Serdyuchenko et al. (2014) | × | × | × | × | √ |
| $O_4$ | 293K, Thalman and Volkamer (2013) | √ | √ | √ | √ | √ |
| HCHO | 298K, Meller and Moortgat (2000) | √ | × | √ | √ | √ |
| Glyoxal | 298K, Volkamer (2005) | × | √ | × | × | × |
| $H_2O$ | HITEMP (Rothman et al. 2010) | √ | √ | √ | √ | × |
| BrO | 223K, Fleischmann et al. (2004) | √ | × | √ | √ | × |
| HONO | 296K, Stutz et al. (2000) | × | × | × | √ | × |
| Ring | Calculated with QDOAS | √ | √ | √ | √ | √ |
| Polynomial degree | | Order 3 | Order 3 | Order 3 | Order 5 | Order 3 |
| Intensity offset | | Constant | Constant | Constant | Constant | No |

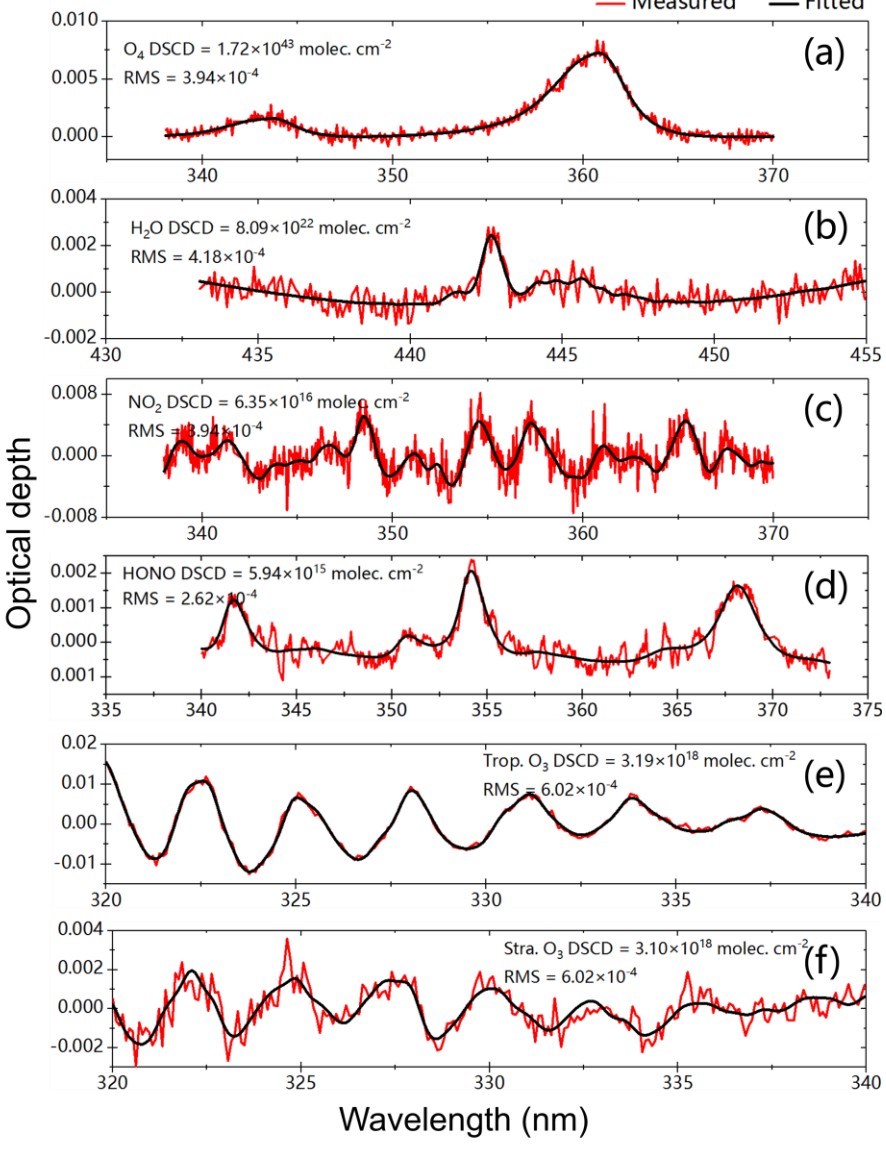

Figure 2. DOAS fit examples of $O_4$, $H_2O$, $NO_2$, HCHO, tropospheric $O_3$ and stratospheric $O_3$. The red line and black line represent the measured and fitted results, respectively.

### 2.2.3 Error analysis

The error sources can be divided into four different types: smoothing error, noise error, forward model error, and model parameter error (Rodgers, 2004). However, in terms of this classification, some errors are difficult to be calculated or estimated. For example, the forward model error, which is caused by an imperfect representation of the physics of the system, is hard to be quantified due to the difficulty of acquiring an improved forward model. Given calculation convenience and contributing ratios of different errors in total error budget, we mainly took into account following error sources, which were smoothing and noise errors, algorithm error, cross section error, and uncertainty related to the aerosol retrieval (only for trace gas). In this study, we estimated the contribution of different error sources to the AOD and VCDs of trace gases, and near-surface (0–200 m) trace gases' concentrations and aerosol extinction coefficients (AECs), respectively. The detailed demonstrations and estimation methods are displayed below.

a.  Smoothing errors arise from the limited vertical resolution of profile retrieval. Noise errors denote the noise in the spectra (i.e., the error of DOAS fits). Considering the error of the retrieved state vector equaling the sum of these two independent errors, we calculated the sum of smoothing and noise errors on near-surface concentrations and column densities, which were 13 and 5 % for aerosols, 13 and 36 % for $H_2O$, 12 and 14 % for $NO_2$, 18 and 21 % for HONO, and 12 and 32 % for $O_3$, respectively.

b.  Algorithm error is denoted by the differences between the measured and simulated DSCDs. This error contains forward model error from an imperfect approximation of forward function, parameter error of forward model, and other errors, such as detector noise (Rodgers, 2004). Algorithm error is a function of the viewing angle, and it is difficult to assign this error to each altitude. Thus, this error on the near-surface values and column densities is estimated through calculating the average relative differences between the measured and simulated DSCDs at the minimum and maximum elevation angle (except 90°), respectively (Wagner et al., 2004). In this study, we estimated these errors on the near-surface values and the column densities at 4 and 8 % for aerosols, 3 and 11 % for $NO_2$, and 20 and 20 % for HONO referring to Wang et al. (2017, 2020), 1 and 8 % for $H_2O$ referring to Lin et al. (2020), and 6 and 10 % for $O_3$ referring to Ji et al. (2023), respectively.

c.  Cross section error arises from the uncertainty in the cross section. According to Thalman and Volkamer, (2013), Lin et al. (2020), Vandaele et al. (1998), Stutz et al. (2000), and Serdyuchenko et al. (2014), we adopted 4, 3, 3, 5, and 2 % for $O_4$ (aerosols), $H_2O$, $NO_2$, HONO and $O_3$, respectively.

d.  The profile retrieval error for trace gases is sourced from the uncertainty of aerosol extinction profile retrieval and propagated to trace gas profile. This error could be roughly estimated based on a linear propagation of the total error budgets of the aerosol retrievals. The errors of the learned four trace gases were roughly estimated at 14 % for VCDs and 10 % for near-surface concentrations, respectively.

The total uncertainty was the sum of all above errors in the Gaussian error propagation, and the error results were listed in Table 2. We found that the smoothing and noise errors played a dominant role in the total uncertainties of aerosol and trace gases. Moreover, improving the accuracy and temperature gradient of the absorption cross section is another important means to reduce the uncertainty of the vertical profiles in the future, especially for $O_3$.

Table 2. Error budget estimation (in %) of the retrieved near-surface (0–200 m) concentrations of trace gases and AECs, and AOD and VCDs.

| | | Error sources | | | | Total |
| | | Smoothing and noise errors | Algorithm error | Cross section error | Related to the aerosol retrieval | |
|---|---|---|---|---|---|---|
| Near-surface | aerosol | 13 | 4 | 4 | / | 14 |
| | $H_2O$ | 13 | 1 | 3 | 14 | 19 |
| | $NO_2$ | 12 | 3 | 3 | 14 | 18 |
| | HONO | 18 | 20 | 5 | 14 | 29 |

| | | 12 | 6 | 2 | 14 | 19 |
|---|---|---|---|---|---|---|
| | $O_3$ | 12 | 6 | 2 | 14 | 19 |
| VCD or AOD | AOD | 5 | 8 | 4 | / | 10 |
| | $H_2O$ | 36 | 8 | 3 | 10 | 38 |
| | $NO_2$ | 14 | 11 | 3 | 10 | 20 |
| | HONO | 21 | 20 | 5 | 10 | 31 |
| | $O_3$ | 32 | 10 | 2 | 10 | 35 |

## 2.3 TUV model

The calculation of photolysis rates of HONO and $O_3$ used TUV radiation model (https://www2.acom.ucar.edu/modeling/tropospheric-ultraviolet-and-visible-tuv-radiation-model) based on a full FORTRAN code. In order to ensure the accuracy of model running, we only selected data in sunny and cloudless days. Moreover, we developed a cloud classification method based on the diurnal variations of Color Index ($CI=I_{330}/I_{360}$) in Figure S2. The initial input parameters were as follows: the AOD at 361 nm was derived from aerosol extinction profiles measured by MAX-DOAS; the daily total ozone column density was measured by TROPOMI with a value range of 260-280 DU; the single scattering albedo (SSA) was calculated based on the regression analysis of multi-wavelength (361 and 477 nm) $O_4$ absorptions measured by MAX-DOAS (Xing et al., 2019); fixed Ångström exponents of 0.508, 0.581 and 0.713 were used in May, June and July, respectively, referring to Xia et al. (2011).

## 2.4 Backward trajectory, PSCF and CWT analysis

The 48-h backward trajectories at five heights of 200, 600, 1000, 1400 and 1800 m were calculated using the Hybrid Single-particle Lagrangian Integrated Trajectory (HYSPLIT) model based on the Global Data Assimilation System (GDAS) to identify the major transport pathways of $O_3$ (Draxler and Hess, 1998). Moreover, the calculated backward trajectories were clustered into three groups using Ward's variance method and Angle Distance algorithm (Ward 1963; Wang et al., 2006).

In order to determine the potential source locations of $O_3$ over CAS (NAMORS), the Potential Source Contribution Function (PSCF) model and Concentration Weighted Trajectory (CWT) model were used (Hong et al., 2019; Ou et al., 2021). The PSCF was calculated through the number of air trajectory endpoints being divided by the number of air trajectory endpoints. Moreover, a weighting function was introduced to reduce the increased uncertainties of PSCF with the increase of the distance between the grid and sampling point. In this study, the set of this weighting function referred to Yin et al. (2017). CWT can be used to calculate the weight concentration through averaging the concentrations associated with trajectories crossing the grid cell. Above weighting function was also introduced to calculate the WCWT (Hsu, et al., 2003). The detailed description of these two models can be found in Wang et al., 2006.

## 2.5 Ancillary data

The surface $NO_2$, HONO and $O_3$ concentrations used to validate the corresponding MAX-DOAS measurements were monitored by broadband cavity enhanced spectrometer (BBCES) (Fang et al., 2017), long path absorption photometer (LOPAP) (Kleffmann et al., 2008) and Thermo Electron 49i (Shi et al., 2009), respectively. The PBL height was simulated using WRF with spatiotemporal resolutions of $20 \times 20$ km$^2$ and 1.0 hour (detailed configurations in Sect. S3 of the supplement). Moreover, the large-scaled spatial distributions of AOD, $O_3$ and $NO_2$ over CAS (NAMORS) were monitored by Himawari-8 (Bessho et al., 2016), OMI (Veefkind et al., 2004) and TROPOMI (Griffin et al., 2018; Su et al., 2020), respectively.

## 3 Results

### 3.1 Overview of the measurements

Figure 3 showed the averaged diurnal variation of AOD from 1[st] May to 9[th] July 2019, with an average value of 0.076 km$^{-1}$ during 08:00-19:00. The AOD was 0.071 km$^{-1}$ at 08:00, and then gradually decreased to a minimum value of 0.052 km$^{-1}$ at 12:00. Subsequently, the AOD increased significantly,

reaching maximum values during 15:00-17:00 (average of 0.107km$^{-1}$), which was about 1.408 times the diurnal average value. Considering the diurnal variation of wind speed (Figure S3), such an enhancement of AOD may be related to the long-range transport of aerosol from southern Asia (Yang et al., 2020; Bi et al., 2023). Moreover, 15:00-17:00 was the active time of tourists and local residents (i.e. cooking), and these kinds of anthropogenic sources contributed to the atmospheric AOD of NAMORS through short-distance transport (Yin et al., 2017; Zhang et al., 2017). After 17:00, the AODs decreased rapidly to 0.071 km$^{-1}$ at 18:00 and 0.081 km$^{-1}$ at 19:00, respectively.

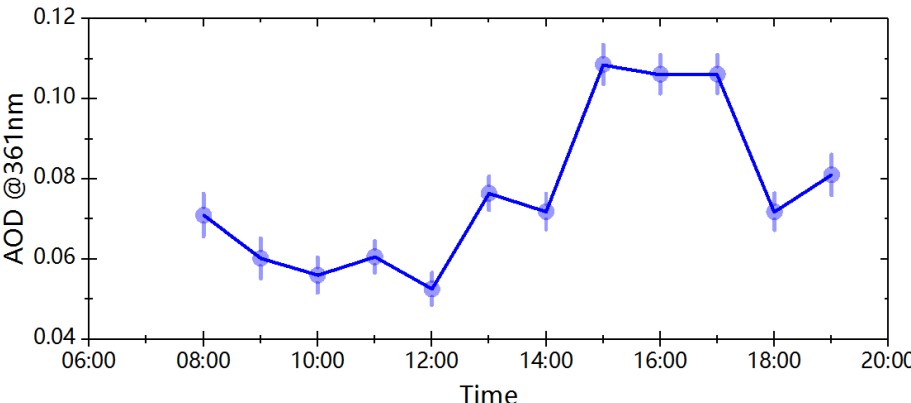

Figure 3. Averaged diurnal variation of AOD at CAS (NAMORS). The error bars represent the mean retrieved errors of AOD.

As shown in Figure S4, the diurnal variation of PBL in Nam Co from May to July 2019 was lower in the early morning and late afternoon, but higher between 11:00 and 17:00, a relatively long period, with the maximum PBL larger than 2.0 km. Zhang et al. (2017) and Yang et al., (2017) also reported that the PBL in Nam Co was usually larger than 1.0 km during daytime in spring and summer. In order to investigate the height-dependent variations of aerosol, $H_2O$, $NO_2$, HONO and $O_3$ within the PBL during the measurements, five height layers under the PBL (0.0-0.2 km, 0.4-0.6 km, 0.8-1.0 km, 1.2-1.4 km and 1.6-1.8 km) were thus selected.

Figure 4 showed the time series of the daily averaged aerosol, $H_2O$, $NO_2$, HONO and $O_3$ at above five layers from 1$^{st}$ May to 9$^{th}$ July 2019. Aerosol mainly distributed at 0.0-0.2 km with an average extinction coefficient of 0.138 km$^{-1}$, and the ratios of aerosol extinction at 0.4-0.6 km, 0.8-1.0 km, 1.2-1.4 km and 1.6-1.8 km to those at 0.0-0.2 km were 39.34%, 18.77%, 7.29% and 2.62%, respectively. That indicated that the aerosol was usually local-emitted at the surface, and the occasionally appearance of strong aerosol extinction at 0.4-0.6 km, such as 13$^{th}$ and 30$^{th}$ June, was associated with long-range transport from south Asia (Figure S5, Wan et al., 2015; Li et al., 2016). The average concentration of $H_2O$ at 0.0-0.2 km was $2.35 \times 10^{17}$ molec cm$^{-3}$, and the ratios of $H_2O$ at 0.4-0.6 km, 0.8-1.0 km, 1.2-1.4 km and 1.6-1.8 km to those at 0.0-0.2 km were 83.40%, 68.08%, 50.64% and 35.74%, respectively, which should attribute to the transport of $H_2O$ from southern Asia driven by the Indian ocean monsoon and the elevated evaporation from Nam Co lake to lead to its not obvious vertical gradient (Figure S6, Lei et al., 2014; Zhu et al., 2019). The average concentration of $NO_2$ at 0.0-0.2 km was 0.193 ppb, and its high concentration mainly distributed at 0.4-0.6 km after 15$^{th}$ May. The ratios of $NO_2$ at 0.4-0.6 km, 0.8-1.0 km, 1.2-1.4 km and 1.6-1.8 km to those at the bottom layer were 104.03%, 59.05%, 24.62% and 12.84%, respectively. The elevation of the distribution height of high concentration $NO_2$ should be attributed to the transport process from the $NO_x$ produced by ice and snow on the top of Mt. Tanggula under strong ultraviolet radiation (Boxe et al., 2005; Fisher 2005; Lin et al., 2021). As depicted in Figure S7, the WPSCF passing through Mt. Tanggula showed high values at 300-400 m layer, especially at 400 m (> 0.3). It also indirectly indicated that the important contribution to $NO_x$ from ice and snow on the top of mountains under strong ultraviolet radiation on the TP. HONO mainly distributed at 0.0-0.2 km with an average value of 0.087 ppb, and the ratios of HONO at 0.4-0.6 km, 0.8-1.0 km, 1.2-1.4 km and 1.6-1.8 km to those at 0.0-0.2 km were 58.49%, 44.64%, 31.30% and 21.67%, respectively. That indicated that the primary and secondary sources of HONO were mainly at the surface (Section 4.2). The vertical gradient of daily averaged $O_3$ concentration was also not obvious, which was associated with its vertical mixing and photochemical

production (Yin et al., 2017). As shown in Figure S8, the corresponding TROPOMI $O_3$ profiles in Nam Co and $O_3$ profiles measured by lidar and ozonesonde around Nam Co reported in several previous studies also exhibited an exponential shape (Fang et al., 2019; Zhang et al., 2020; Yu et al., 2022). The $O_3$ average concentration at 0.0-0.2 km was 63.030 ppb, and the ratios of $O_3$ at 0.4-0.6 km, 0.8-1.0 km, 1.2-1.4 km and 1.6-1.8 km to those at surface were 89.25%, 82.44%, 80.16% and 79.13%, respectively.

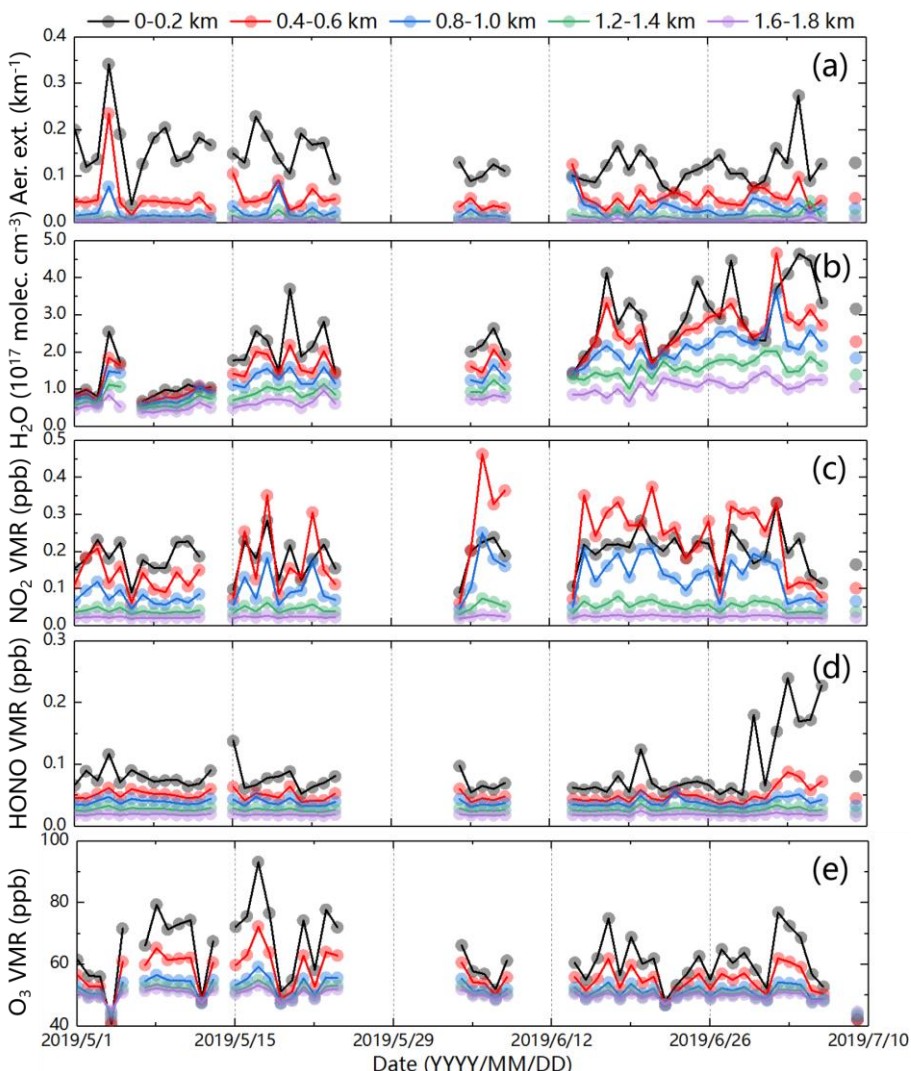

Figure 4. Time series of daily averaged (a) aerosol extinction, (b) $H_2O$, (c) $NO_2$, (d) HONO, and (e) $O_3$ monitored by MAX-DOAS at 0-0.2, 0.4-0.6, 0.8-1.0, 1.2-1.4 and 1.6-1.8 km five height layers from 01 May to 09 July 2019.

**3.2 Vertical distributions of aerosol, $H_2O$, $NO_2$, HONO and $O_3$**

The first row in Figure 5 provided the averaged vertical profiles of aerosol, $H_2O$, $NO_2$, HONO and $O_3$ from May to July 2019. We found that the vertical profiles of aerosol, $H_2O$, HONO and $O_3$ all exhibited an exponential shape with maximum values near the surface, while $NO_2$ exhibited a Gaussian shape with the maximum value of 0.321 ppb occurring at 0.3-0.4 km layer. In addition to the effect of $NO_x$ transport, Xu et al. (2018) also revealed that the long-range high-altitude transport process from the northern south Asian subcontinent can significantly enhance the Nam Co's peroxyacetyl nitrate (PAN) level which is a reservoir of $NO_x$. As shown in the second row of Figure 5, the monthly averaged aerosol vertical profiles from May to July 2019 all exhibited an exponential shape, and varied in the order of May (0.17 $km^{-1}$) > July (0.14 $km^{-1}$) > June (0.11 $km^{-1}$). Xu et al. (2018) and Neupane et al. (2019) also reported a similar monthly variations of black carbon (BC) from May to July over the TP, and revealed that it was mainly associated with the anthropogenic emissions (i.e. biomass burning) and its transport from south Asia. The monthly averaged vertical profile of $H_2O$ in May and July exhibited an exponential shape, while its maximum concentration layer slightly elevated to 0.1-0.2 km

in June which was related to the strongest monsoon transport (Figure S9). It varied in the order of July
$(3.68 \times 10^{17}$ molec cm$^{-3}) >$ June $(2.71 \times 10^{17}$ molec cm$^{-3}) >$ May $(2.26 \times 10^{17}$ molec cm$^{-3})$, and its
maximum concentration occurring in July was strongly associated with the enhanced evaporation from
the Nam Co lake (Xu et al., 2011). The monthly averaged vertical profiles of $NO_2$ all exhibited a
Gaussian shape from May to July, and its maximum values mainly distributed at 0.2-0.4 km layer
varying in the order of June (0.39 ppb) > May (0.31 ppb) > July (0.28 ppb). It indicated that the
regional transport from the $NO_x$ produced from ice and snow under strong shortwave radiation (Figure
S7), $NO_2$ emitted from vehicles due to the increased tourism, anthropogenic emissions from local
residents (i.e. biomass burning and religious activities) played an important role in the vertical
distribution characteristic of $NO_2$ (Boxe et al., 2005; Chen et al., 2019). The monthly averaged vertical
profiles of HONO from May to July all exhibited an exponential shape, with maximum values near the
surface varying in the order of July (0.13 ppb) > May (0.07 ppb) > June (0.06 ppb). The local direct
emissions from biomass burning, vehicles and soil should be main sources of the surface HONO (Xing
et al., 2021). Moreover, the heterogeneous reaction of $NO_2$ on wet surfaces should be another important
source of HONO at different height layers (Section 4.2). For example, the aerosol extinction coefficient,
and the concentrations of $H_2O$ and $NO_2$ were all relatively large at the bottom layer in July,
correspondingly, we observed the highest concentration of HONO near the surface in this month. The
monthly averaged $O_3$ vertical profiles all showed an exponential shape from May to July, and its
surface concentration varied in the order of May (66.71 ppb) > July (61.45 ppb) > June (59.55 ppb).
This kind of monthly variation trend of $O_3$ was also reported by several previous studies (Yin et al.,
2017; Xu et al., 2018). The $O_3$ in Nam Co was mainly sourced from stratospheric intrusion,
photochemical reactions, long-range transport and local vertical mixing (Yin et al., 2017; Chen et al.,
341  2019).

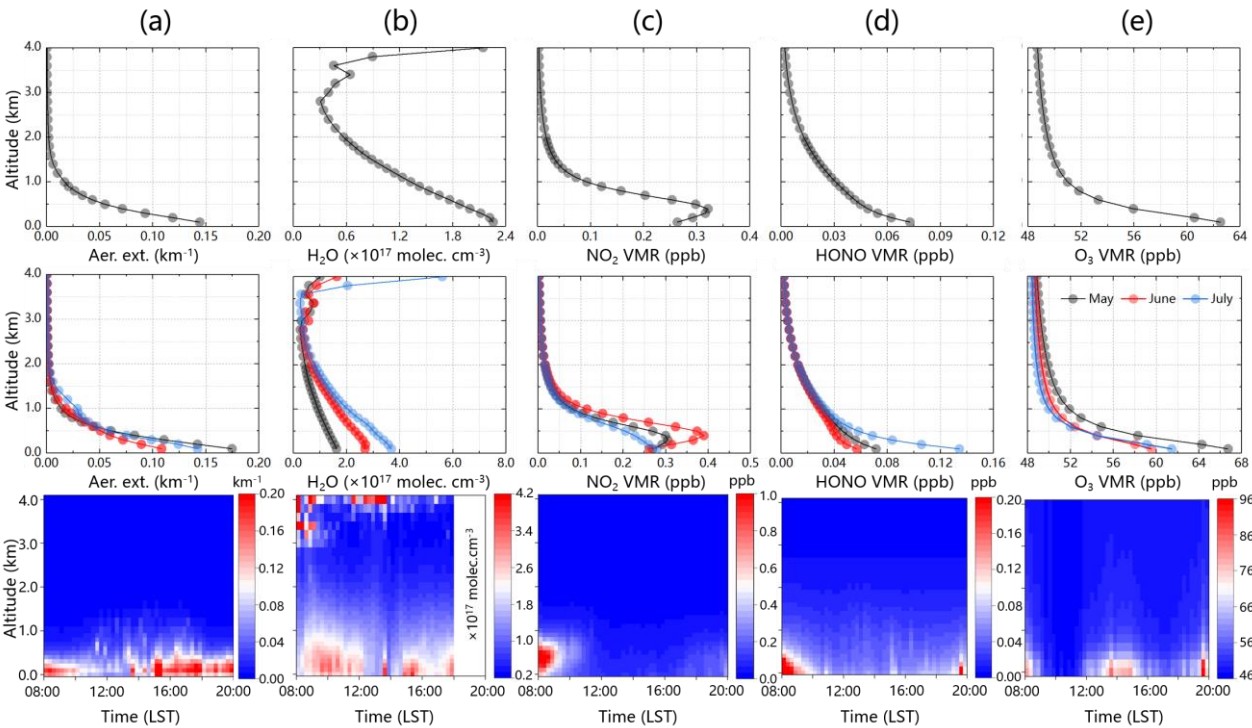


Figure 5. Vertical profiles of (a) aerosol extinction, (b) $H_2O$, (c) $NO_2$, (d) HONO, and (e) $O_3$. The top
row shows the averaged vertical profiles from 01 May to 09 July 2019. The middle row shows the
monthly averaged vertical profiles. The bottom row shows the averaged diurnal vertical profiles from
01 May to 09 July 2019.
The third row in Figure 5 illustrated the averaged diurnal variations in vertical profiles of aerosol, $H_2O$,
$NO_2$, HONO and $O_3$ from May to July 2019. Aerosol mainly distributed under 1.0 km, especially 0.6
km, and its mixing height was gradually increased with the rise of the PBL height after 12:00.
Moreover, the diurnal variation of aerosol showed a bi-peak pattern, which was in line with the
investigation reported by Pokharel et al. (2019). The first peak occurred between 08:00-10:00, and

another appeared after 15:00. The first peak should be attributed to the local emission of aerosol and the diurnal cycle of PBL (Zhang et al., 2017; Pokharel et al., 2019). The second peak was driven by regional transport and the interaction between local sandy silt loam surface and local meteorology. The high wind speed (> 4.5 m/s) at surface appeared after 15:00, which coincided with the appearance of the second aerosol peak (Figure S3). Moreover, the high extinction during the second peak was extended to 1.0 km associated with the wind speed larger than 8 m/s (Figure S10), which created a favorable condition for high-altitude aerosol transport. $H_2O$ mainly distributed under 1.0 km and above 3.0 km, and its diurnal variation exhibited a multi-peak pattern. The first peak appeared between 08:00-12:00, which was mainly affected by the monsoon driven long-range transport of $H_2O$ (Cong et al., 2009; Xu et al., 2020). The second and third peaks occurred at 15:00-16:00 and after 17:00, respectively. In addition to long-range transport, the enhanced evaporation from the Nam Co lake also significantly contributed to the appearance of these two peaks of $H_2O$ (Xu et al., 2011). $NO_2$ mainly distributed at 0.2-0.4 km, and peaked before 10:00 and after 18:00 which were dominated by the effects of local emissions and regional transport from the NOx formed through ice and snow on the top of Mt. Tanggula under strong ultraviolet radiation (Figure S7) (Boxe et al., 2005; Fisher 2005; Chen et al., 2019; Lin et al., 2021). Moreover, its diurnal mixing height was obviously correlated to the diurnal evolution of PBL height. HONO mainly distributed under 1.0 km, especially 0.4 km. Its diurnal variation showed a multi-peak pattern with three obvious peaks before 10:00, 15:00-16:00, and after 19:00. In addition to local emissions (i.e. vehicle emission, biomass burning and soil emission), the heterogeneous reaction of $NO_2$ on wet surfaces should be also an important HONO source (Xing et al., 2021). We found that there were larger aerosol extinction (> 0.12 km$^{-1}$) and higher concentrations of $NO_2$ (> 0.20 ppb) and $H_2O$ (> $2.27 \times 10^{17}$ molec cm$^{-3}$) around three HONO peaks. $O_3$ mainly distributed under 0.4 km, and its diurnal variation exhibited a multi-peak pattern with three peaks appearing before 09:00, 13:00-15:00 and after 19:00. The appearance of $O_3$ peaks was mainly associated with the influence of the complex topography of the TP, long-range transport, local vertical mixing and stratospheric intrusion (Yin et al., 2017; Chen et al., 2019; Qian et al., 2022). The active photochemical reaction should be another important source of $O_3$, especially for its second peak at 13:00-15:00.

### 3.3 Validation with independent data

In order to validate the MAX-DOAS dataset, we extracted the concentrations of $NO_2$, HONO and $O_3$ at the bottom layer (0.0-0.1 km) from their corresponding vertical profiles to compare with in situ measurements. As shown in Figure 6(a-c), we found good agreements between MAX-DOAS and in situ observations with Pearson correlation coefficients (R) of 0.91, 0.62 and 0.82 (regression slope of 0.89, 1.05 and 0.82) for $NO_2$, HONO and $O_3$, respectively. That indicated the good reliability of trace gases from MAX-DOAS retrievals. Moreover, we also compared the MAX-DOAS PBL and WRF PBL, and a similar variation trend was found. However, WRF PBL showed a significantly difference in height values with MAX-DOAS PBL before 12:00. That should be due to the simulation uncertainties for WRF model at Tibetan plateau with complex topography and meteorology (Yang et al., 2016; Xu et al., 2019).

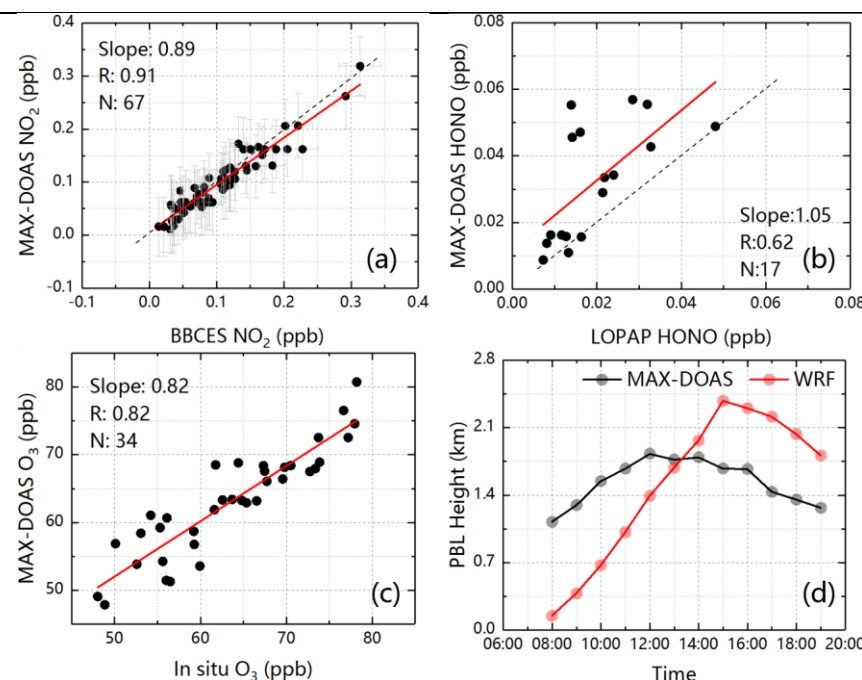


Figure 6. Validations of (a) MAX-DOAS $NO_2$ vs in situ $NO_2$ (error bars represent the retrieved errors
of $NO_2$ from MAX-DOAS and BBCES), (b) MAX-DOAS HONO vs LOPAP HONO, (c)
MAX-DOAS $O_3$ vs in situ $O_3$, and (d) MAX-DOAS PBL vs WRF PBL.

## 4 Discussion

### 4.1 OH production

HONO and $O_3$ are two important precursors of OH redical to enhance the AOC (Kleffmann et al., 2005;
Ryan et al., 2018; Xing et al., 2021). In order to evaluate the AOC on the TP, we tried to analyze the
OH production from HONO and $O_3$ at different height layers through vertical observations and TUV
calculations. The OH production rates from HONO and $O_3$ were calculated using the following two
equations:

$$P(OH)_{HONO} = J(HONO) \times [HONO]$$

$$P(OH)_{O_3} = 2 \times f \times J(O(^1D)) \times [O_3]$$

Where *J(HONO)* and *J(O($^1$D))* were the photolysis rates of HONO and O($^1$D) calculated using TUV
model. O($^1$D) was the product from $O_3$ photolysis by UV radiation. *f* was the fraction of the process
O($^1$D) + $H_2O$ → 2OH.
Figure 7(a-b) showed the averaged diurnal vertical distributions of the photolysis rates *J(HONO)* and
*J(O($^1$D))* from May to July 2019. We found that the maximum *J(HONO)* and *J(O($^1$D))* were all
appeared at the bottom layer between 12:30 and 15:30 with values of $2.0 \times 10^{-3}$ and $6.75 \times 10^{-5}$ $s^{-1}$,
respectively. The maximum values were usually larger than that at low-altitude areas due to the
stronger solar UV radiation on the TP (Su et al., 2008; Xing et al., 2021; Yang et al., 2021; Liu et al.,
2022), but being consistent with the values on the TP reported by Lin et al. (2008). Moreover, it should
be noted that the values of *J(HONO)* and *J(O($^1$D))* all decreased with the increase of altitude, which
was significantly different with previous studies in low altitudes (Ryan et al., 2018; Xing et al., 2021;
Xu et al., 2021).

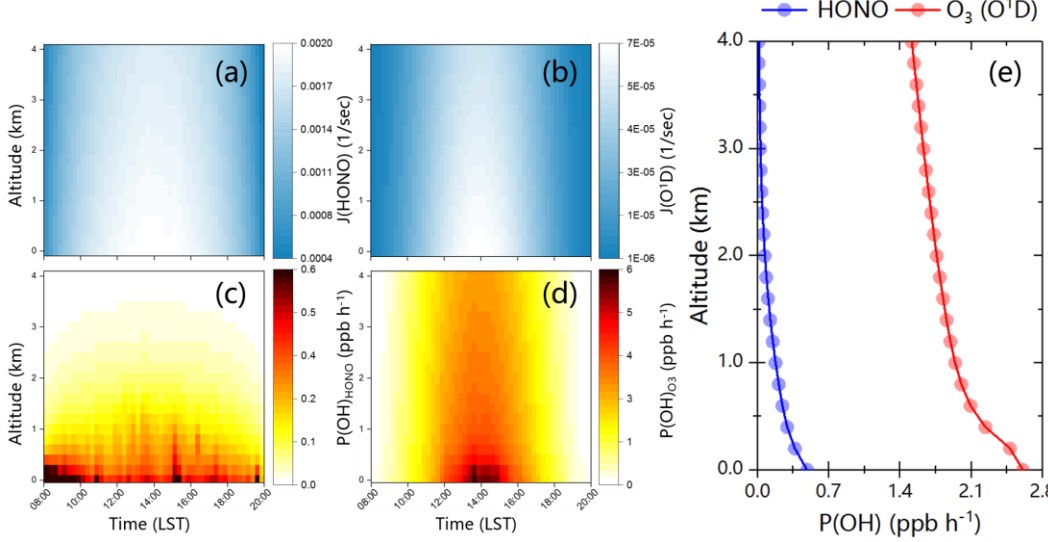


Figure 7. Averaged diurnal vertical profiles of the (a) photolysis rate *J(HONO)*, (b) photolysis rate
*J(O($^1$D))*, (c) OH radical production rates from HONO photolysis, (d) OH radical production rates from
O$_3$ photolysis. (e) shows the averaged vertical profiles of OH radical production rates from HONO and
O$_3$ photolysis from 01 May to 09 July 2019.
Figure 7(c-d) showed the averaged diurnal vertical profiles of OH production rates from HONO and O$_3$
photolysis from May to July 2019. P(OH)$_{HONO}$ exhibited a multi-peak pattern which mainly appeared
before 10:00, 15:00-16:00, and after 19:00 at 0-0.4 km with a maximum value of 0.81 ppb/h. While
P(OH)O$_3$ showed a unimodal pattern occurring at 13:00-15:00 under 0.4 km with a maximum value of
6.20 ppb/h. The averaged vertical profiles of P(OH)$_{HONO}$ and P(OH)$_{O3}$ during the observation were
depicted in Figure 7(e). We found that the maximum values of P(OH)$_{HONO}$ (0.49 ppb/h) and P(OH)$_{O3}$
(2.61 ppb/h) all appeared at the bottom layer, and decreased with height. That indicated O$_3$ was an
important contributor of OH production (> 80%) on the TP, which was about 5-6 times to HONO.
Moreover, the OH production rates from HONO and O$_3$ in other cities of China were depicted in Table
3. The contribution percentage of O$_3$ to P(OH) in Nam Co was significantly higher than that in other
cities, which was due to the relatively high concentrations of O$_3$ and H$_2$O, and the strong radiation in
Nam Co. In addition, P(OH)$_{HONO}$ in Nam Co was close to that in relatively dry areas (i.e. Beijing and
Xianghe), but slightly lower than that in areas with relatively high humidity which can enhance the
heterogeneous production of HONO (Ryan et al., 2018; Liu et al., 2019; Xing et al., 2021).
Table 3. The maximum OH production rates contributed from HONO and O$_3$ at different locations.

| Location | Date | *P(OH)$_{HONO}$* (ppb/h) | *P(OH)$_{O3}$* (ppb/h) | References |
|---|---|---|---|---|
| Xianghe (China) | Jul. 2008-Apr. 2009 | ~0.80 in Spring ~0.70 in Summer | ~0.20 in Spring, ~0.45 in Summer | Hendrick et al. (2014) |
| Beijing (China) | Mar. 2010-Dec. 2012 | ~1.25 in Spring, ~0.70 in Summer | ~0.10 in Spring, ~0.55 in Summer | Hendrick et al. (2014) |
| East China Sea (China) | Jun. 2017 | ~1.75 | ~1.20 | Cui et al. (2019) |
| Chengdu (China) | Aug.-Sep. 2019 | ~3.25 | - | Yang et al. (2021) |
| Qingdao (China) | Jul.-Aug. 2019 | ~1.30 | ~1.00 | Yang et al. (2021) |
| Nam Co (China) | May-Jul. 2019 | 0.81 | 6.20 | This study |

**4.2 Possible daytime HONO sources**
Atmospheric HONO mainly sourced from direct emission, homogeneous reaction and heterogeneous
reaction (Fu et al., 2019; Ren et al., 2020; Chai et al., 2021; Crilley et al., 2021; Li et al., 2021). There
were less anthropogenic emissions for HONO around NAMORS, however, the open burning of crop
residues and soil emissions should be important HONO sources considering the pasture environment
and large amounts of animal manure (Cui et al., 2021a; 2021b). Moreover, the background of low-level
NO on the TP leaded to the homogeneous reaction not to be the main source of HONO at NAMORS
(Lin et al., 2019; Xing et al., 2021; Li et al., 2022). Heterogeneous reaction of NO$_2$ on wet surfaces
became an important potential source of HONO around NAMORS, which affected by the humidity,

temperature, solar radiation, aerosol concentration and corresponding specific surface area. In order to remove the effect of diurnal PBL evolution, we used $HONO/NO_2$ to indicate the extent of the heterogeneous reaction process. As shown in Figure 8, scatter plots between $HONO/NO_2$ and $H_2O$ were illustrated. We found that the maximum value of $HONO/NO_2$ appeared around water vapor being around $1.0 \times 10^{17}$ molec $cm^{-3}$ under 1.0 km, and being around $0.5-1.0 \times 10^{17}$ molec $cm^{-3}$ at 1.0-2.0 km height layer. This phenomenon of $HONO/NO_2$ firstly increasing and then decreasing with the increasing of $H_2O$ (or relative humidity) was usually found in low-altitude areas in previous studies (Wang et al., 2013; Liu et al., 2019; Xing et al., 2021; Xu et al., 2021). When the $H_2O$ was greater than above mentioned critical values at different heights, $HONO/NO_2$ gradually decreased, which was related to the efficient uptake of HONO and the decrease of $NO_2$ reactivity with the increase of $H_2O$ (Liu et al., 2019; Xu et al., 2021). That indicated $H_2O$ has significant enhancement for the conversion rate of $NO_2$ to HONO. Moreover, we found that the high value areas of $HONO/NO_2$ at above five height layers were all accompanied by high aerosol extinction ($> 0.15$ $km^{-1}$ under 1.0 km, and $> 0.02$ $km^{-1}$ at 1.0-2.0 km). It indicated that aerosol surface has contribution to the heterogeneous reaction process of $NO_2$. The scatter plots between HONO and $NO_2$ at above five layers (Figure S11) also confirmed the possibility of the $NO_2$ heterogeneous reaction to generate HONO on the TP, and the contribution of atmospheric $H_2O$ and aerosol extinction to this process.

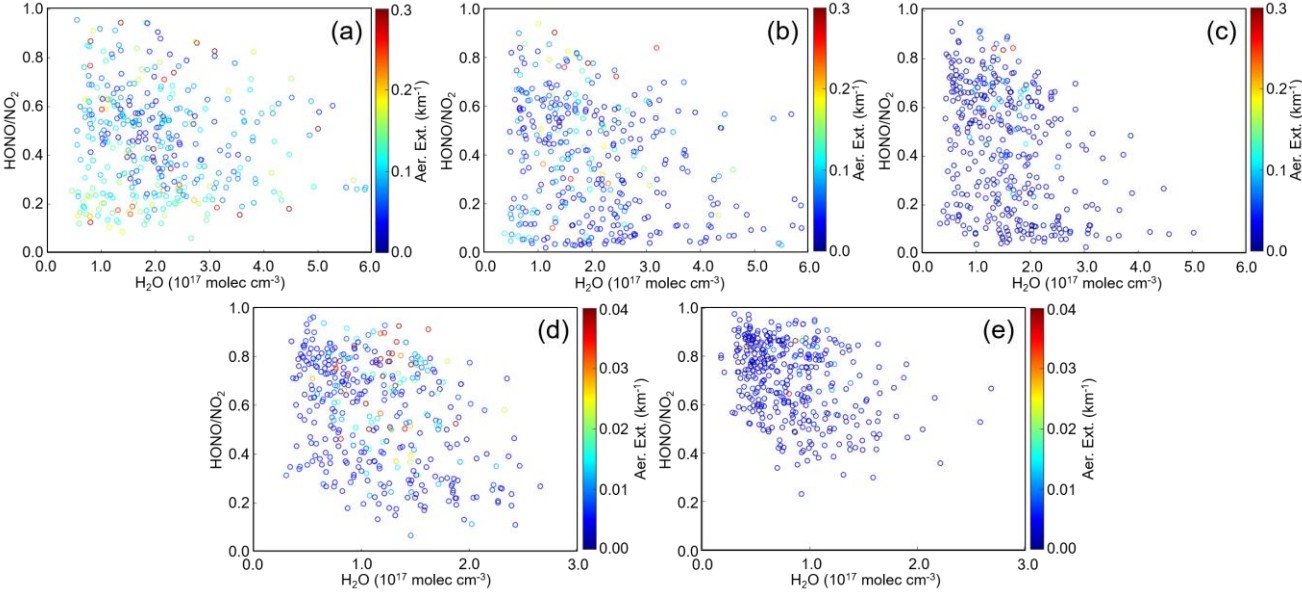

Figure 8. Scatter plots between $HONO/NO_2$ and $H_2O$ colored by aerosol extinction at (a) 0.0-0.2 km, (b) 0.4-0.6 km, (c) 0.8-1.0 km, (d) 1.2-1.4 km, (and e) 1.6-1.8 km from 1st May to 9th July 2019.

In Figure 9, the vertical profile of $HONO/NO_2$ from May to July 2019 was depicted. We found that $HONO/NO_2$ firstly decreased and then increased with the increasing of height, which was opposite to previous studies in low-altitude areas (Meng et al., 2020; Zhang et al., 2020; Xing et al., 2021; Xu et al., 2021). The minimum average $HONO/NO_2$ occurred at 0.3-0.4 km height layer with a value of 0.37. The relatively high values of $HONO/NO_2$ at the bottom layer should be related to the non-deducted HONO direct emissions.

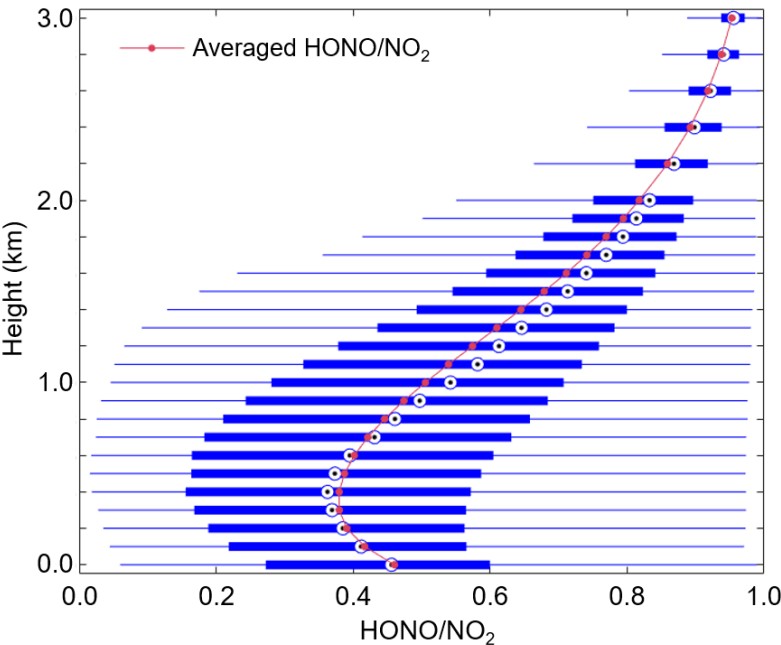

471

Figure 9. Statistics for the vertical profile of HONO/NO$_2$ from 1$^{st}$ May to 9$^{th}$ July 2019. The left and right of the blue box represent the 25$^{th}$ and 75$^{th}$ percentiles, respectively; the dot within the box represents the mean.

### 4.3 Possible daytime O$_3$ sources

In addition to local photochemistry process, long-range transport was the main source of O$_3$ on the TP (Yin et al., 2017; Xu et al., 2018). To further understand the transport pathway and potential source of O$_3$, cluster analysis, WPSCF and WCWT models were used to assess the regional representativity of O$_3$ at five typical heights (200 m, 600 m, 1000 m, 1400 m and 1800 m). As shown in Figure S12 and Table 4, the backward trajectories arriving at NAMORS during the observation were classified into three clusters at 200 m, 600 m, 1400 m, 1800 m, and four clusters at 1000 m. We found that cluster 3 was associated with the highest O$_3$ concentration at 200 m (65.48$\pm$17.41 ppb) and 1800 m (49.69$\pm$ 2.21 ppb), and cluster 1 were related to the highest O$_3$ concentration at 600 m (54.67$\pm$6.94 ppb), 1000 m (51.61$\pm$3.84 ppb) and 1400 m (50.51$\pm$2.89 ppb). These two clusters were all originating from northwestern of south Asian subcontinent passing through Himalayas, which was also reported by Yin et al. (2017) during springtime from 2011 to 2015. In Figure S13 and 10, WPSCF and WCWT analysis told us that the high O$_3$ concentration at above heights potentially sourced from northern India, central Pakistan, Nepal, western Bhutan and northern Bangladesh through long-range transport. It should be noted that the potential contribution to O$_3$ at NAMORS at 200 m from above potential source areas were all over 40 ppb. These contributions from the mentioned potential source areas at other four heights were also over 20-30 ppb. The massive fire emissions during springtime were an important source of O$_3$ in south Asia (Jena et al., 2015), and the obvious burning during the observation was observed in Figure S14. Moreover, the abundant precursors and high photochemical activity were another significant sources of O$_3$ in south Asia (Kumar et al., 2012; Sharma et al., 2017).

In addition, Figure 10 showed that the contribution of O$_3$ transported from Himalayas can even up to 50 ppb, especially under 600 m. Several previous studies have revealed that the stratospheric O$_3$ intrusion events were frequent in the Himalayas during spring and summer (Cristofanelli et al., 2010; Chen et al., 2011; Škerlak et al., 2014; Putero et al., 2016). Therefore, the O$_3$ from stratospheric intrusions in the Himalayas can affect the O$_3$ at NAMORS through long-range transport.

Table 4. Trajectory ratios and averaged O$_3$ concentration for all trajectory clusters arriving in Nam Co at 200 m, 600 m, 1000 m, 1400 m and 1800 m from May to July 2019.

| | Cluster | Traj_ratio | O$_3$ concentration (ppb) |
|---|---|---|---|
| | | | Mean$\pm$SD |

| | | | |
|---|---|---|---|
| 200 m | 1 | 55.86% | 61.50±18.15 |
| | 2 | 11.85% | 54.57±14.67 |
| | 3 | 32.28% | 65.48±17.41 |
| | All | 100.00% | 61.14±17.74 |
| 600 m | 1 | 62.55% | 54.67±6.94 |
| | 2 | 14.32% | 50.43±6.64 |
| | 3 | 23.13% | 53.27±7.63 |
| | All | 100.00% | 53.39±7.26 |
| 1000 m | 1 | 49.16% | 51.61±3.84 |
| | 2 | 8.81% | 49.60±3.99 |
| | 3 | 22.73% | 50.72±4.21 |
| | 4 | 19.30% | 51.39±4.49 |
| | All | 100.00% | 50.98±4.30 |
| 1400 m | 1 | 80.14% | 50.51±2.89 |
| | 2 | 4.95% | 49.12±2.73 |
| | 3 | 14.92% | 49.44±3.85 |
| | All | 100.00% | 50.07±3.15 |
| 1800 m | 1 | 83.75% | 49.68±2.55 |
| | 2 | 0.00% | 49.07±2.23 |
| | 3 | 16.25% | 49.69±2.21 |
| | All | 100.00% | 49.59±2.49 |

502

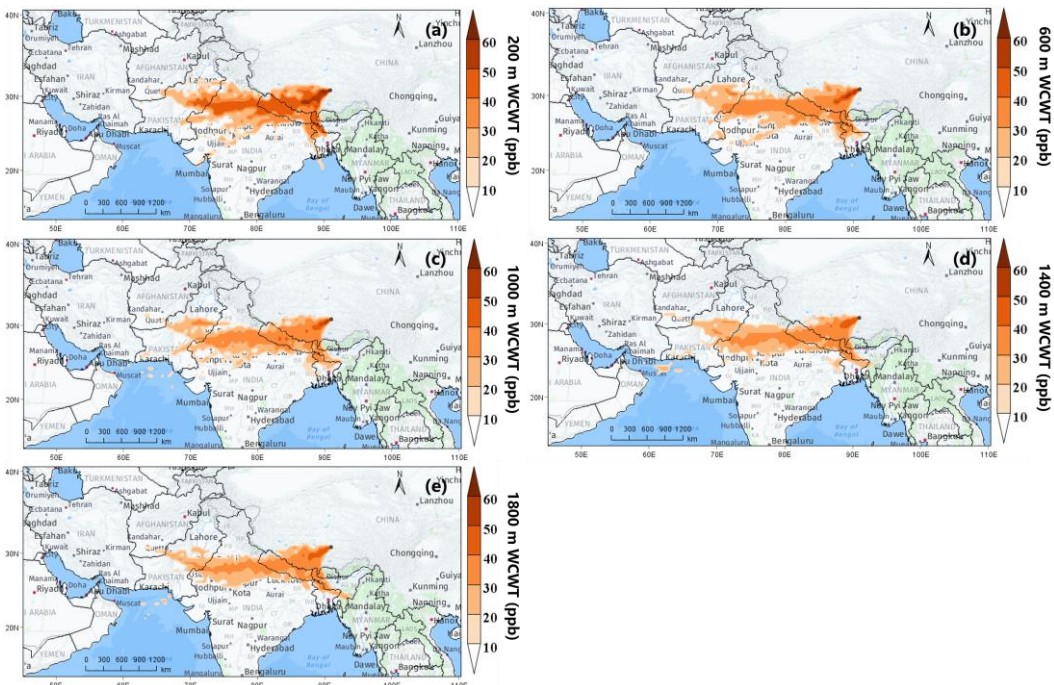

503

Figure 10. Spatial distributions of WCWT values for $O_3$ at (a) 200 m, (b) 600 m, (c) 1000 m, (d) 1400 m, and (e) 1800 m height layers from 01[st] May to 09[th] July 2019 over CAS (NAMORS).

## 5 Summary and conclusions

MAX-DOAS measurements were performed to clarify the vertical distributions of several atmospheric components (aerosol, $H_2O$, $NO_2$, HONO and $O_3$), and to explore the AOC in vertical space in Nam Co from May to July 2019. The MAX-DOAS $NO_2$, HONO and $O_3$ agreed well with in situ measurements, with correlation coefficients of 0.91, 0.62 and 0.82, respectively. We found that the averaged vertical

profiles of aerosol, $H_2O$, HONO and $O_3$ all exhibited an exponential shape, while $NO_2$ showed a Gaussian shape with a maximum value of 0.32 ppb appearing at 300-400 m. The maximum concentrations of monthly averaged aerosol (0.17 $km^{-1}$) and $O_3$ (66.71 ppb) appeared on May, $H_2O$ ($3.68\times10^{17}$ molec $cm^{-3}$) and HONO (0.13 ppb) appeared on July, and $NO_2$ (0.39 ppb) occurred on June. For the diurnal variation, above five species all mainly distributed under 1.0 km, and mostly exhibited a multi-peak pattern considering the effect of regional transport and local chemical reaction.

$O_3$ and HONO were important source of OH on the TP. The diurnal averaged OH production rate from HONO during the observation exhibited a multi-peak pattern appearing before 10:00, 15:00-16:00 and after 19:00 under 0.4 km with the maximum value of 0.81 ppb/h. The OH production rate from $O_3$ shown a unimodal pattern occurring at 13:00-15:00 under 0.4 km with the maximum value of 6.20 ppb/h which was obviously higher than that at low-altitude areas. In addition to direct emission, the heterogeneous reaction of $NO_2$ on wet surfaces was also an important source of HONO in Nam Co. We found that $HONO/NO_2$ first increasing and then decreasing with the increasing of $H_2O$. The maximum value of $HONO/NO_2$ appeared around $H_2O$ being around $1.0\times10^{17}$ molec $cm^{-3}$ under 1.0 km, and being around $1.0\text{-}2.0\times10^{17}$ molec $cm^{-3}$ at 1.0-2.0 km height layer. Moreover, high values of $HONO/NO_2$ usually accompanied by high aerosol extinction. $O_3$ under 2.0 km were potentially sourced from Himalayas, northern India, central Pakistan, Nepal, western Bhutan and northern Bangladesh through long-range transport. Our results draw a picture of further understanding the spatial and temporal variations in oxidation chemistry under PBL and provided a new perspective for source analysis of major atmospheric components through vertical observation on the TP.

## Acknowledgements

We firstly would like to thank @Tibet group for effectively organizing the Nam Co observation. We also would like to thank Peking University (Chunxiang Ye's group) and Anhui Institute of Optics and Fine Mechanics (Weixiong Zhao's group) to provide the DOAS validation data of HONO, $O_3$ and $NO_2$. We thank the National Oceanic and Atmospheric Administration (NOAA) Air Resources Laboratory (ARL) for providing the open HYSPLIT transport and dispersion model. This study was supported by the National Natural Science Foundation of China (42225504 and U21A2027), the Anhui Provincial Natural Science Foundation (2108085QD180), and the Presidential Foundation of the Hefei Institutes of Physical Science, Chinese Academy Sciences (YZJJ2021QN06).

## Competing interests

All authors declare that they have no conflict of interest or financial conflicts to disclose.

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
