# Peer review of "Observations of the vertical distributions of summertime"

_EGUsphere, 2024_

## Referee Comment (RC2)

This manuscript describes ground-based MAX-DOAS measurements in Nam Co over the Tibetan Plateau (TP). The measurements are used to obtain vertical distributions of several atmospheric components (aerosol, H2O, NO2, HONO and O3) via spectral analysis and OEM profile retrieval. The data are further used to analyze the temporal and vertical evolutions for these species. The OH production as well as possible daytime HONO and O3 sources were also discussed during the field campaign. Overall, observing and investigating the vertical profiles of atmospheric components at the background station in the relatively remote and high-altitude region are significant. However, two concerns at least should be clarified in detail before the manuscript is considered to be accepted.

(1) Firstly, the reliabilities of vertical profile of atmospheric components (aerosol, H2O, NO2, HONO and O3) should be validated. Just using the parameter setting scheme of spectral analysis and profile retrieval in previous studies won't do for the specific station over the TP. For example, the HONO spectral structures are almost drowned in the noise in Figure 2d. The sensitivity tests about the parameter setting scheme of spectral analysis and profile retrieval should be presented in detail. In addition, just according to the comparison of surface multi-source data in Section 3.3, it cannot illustrate the reliabilities of vertical profiles. Also, the correlation was weak for HONO ($R^2$=0.38) with larger deviations in Figure 6b.

(2) The manuscript tried to analyze the OH production (Section 4.1) from HONO and O3 at different height layers through vertical observations and TUV calculations. The TUV model is suitable for exploring the photolysis rates, but OH production is determined by complex chemical process involving many atmospheric components. The equations in Line 342 and 343 are simplified for studying OH production. It should present more details on the parameter scheme during simulation by TUV model, and more comparisons of OH production simulated by other models if possible.

**specific comments**

1. Authors' affiliations and addresses should be numbered in the appearance order of their names, to avoid the misunderstanding of academic misconduct.

2. Line 4: The full name should be given when the abbreviation ("MAX-DOAS") first appears. Please modify similar problems elsewhere (For example, "TUV" in Line 100).

3. Line 27 and Line 575: The references' format is not consistent between the text and the reference section, which leads to poor readability of the manuscript for review. Please modify all the similar problems.

4. Line 26-38: This paragraph is not closely related to the key ideas of this paper. Moreover, the last sentence is lack of proper deduction.

5. Line 39-73: This paragraph is lack of many key references for MAX-DOAS observations over the Tibetan Plateau. It is not appropriate to cite only the studies from author's group.

6. Line 75: The $O_3$ will be diluted when the air mass comes from clean source regions, such as marine atmosphere.

7. Line 122: Again, sensitivity test and profile validation by independent data should be presented to confirm the reliability of retrieval results.

8. Line 156: The cross sections are lack of references in Table 1.

9. Line 165: Are only the HONO and O3 data in clear sky condition used to calculate the photolysis rates. If so, there may be many missing data in summer due to clouds over the TP. This information should be explicitly mentioned in section 4.1.

10. Line 177: Please add references.

11. Line 204-207:With respect to the enhancement of AOD during 15:00-17:00, the explanations are far-fetched. The daytime AOD diurnal variations at other time are probably affected by the long-range transport of aerosol and local anthropogenic sources (such as cooking in the morning).

12. Line 215: The manuscript selected 5 height layer to analyze temporal and vertical variations of atmospheric components. But why are the five height layers "typical"?   The separate layers reduce the vertical resolution of retrieval results. It is better to investigate the daily variations through vertical profiles themselves. Please modify all the similar problems at other sections.

13. Line 224-225, 228-229: The explanations are lack of proper deduction.

14. Line 233-234: As a short-life atmospheric component, can the $NO_X$ around Mt. Tanggula transport to the observation site ? Does the transport effect affect the $NO_2$ at the bottom layer ? The causes of elevated $NO_2$ layer are probably complex.

15. Line 240-243: The vertical distributions of $O_3$ in the lower tropospheric layer are complex over the Tibetan Plateau. Why does the manuscript describe the "exponential shape" derived from previous studies here ? The "exponential shape" is not consistent with the relatively uniform vertical variation of $O_3$ in this study.

16. Line 251: Please add "$O_3$".

17. Line 254-255: Why are $NO_2$ vertical profiles "Gaussian" distribution rather than Lorentz or other peak distribution ?

18. Line 264-265: Please clarify the monsoon transport leading to the elevation of maximum $H_2O$ layer.

19. Line 279-282: With respect to the vertical profiles of atmospheric component at the height of 3-4 km, are the retrieved data valid ? How to understand the differences of $O_3$ profiles between "exponential shape"(Line 282) and "relatively uniform vertical gradient" (Line 240) ?

20. Line 298-301: It just lists the possible influencing factors here. To what extent, can the weak surface wind affect the sand-raising process ?

21. Line 302-308: Please check the attribution of $H_2O$ and $NO_2$ variations. For example, the monsoon not only affect the $H_2O$ variation in the morning but also in the afternoon.

22. Line 314-315: Please check that the text description is consistent with the figures.

23. Figure 7: The validity of results above 3 km should be carefully checked. Probably the concentrations of atmospheric component at higher height mainly reflected a priori information.

24. Line 388-390: Is that so ?

25. Figure 9: Please add the graphic symbol description, such as error bar. Please modify all the similar problems in other figures.

26. Line 420: three clusters ?    There are four clusters in Figure S7e.

27. Line 434-437: The discussion of stratospheric $O_3$ intrusion is too simple. The logic is confusing between stratosphere intrusion and long-range transport.

28. Section References (Line 482-854): Although there are too many references, the key references related to this topic are missing.

---

## Author Comment (AC1)

Point-to-point responses

*We appreciate the reviewers for their valuable and constructive comments, which are very helpful for the improvement of the manuscript. We have revised the manuscript carefully according to the reviewers' comments. We have addressed the reviewers' comments on a point-to-point basis as below for consideration, where the reviewers' comments are cited in **black**, and the responses are in **blue**.*

**Referee #1**

Xing et al. learned the vertical variations and sources of $O_3$ and HONO and their precursors on the Tibetan Plateau. The authors found that the contributions of HONO and $O_3$ to the production rates of OH on the TP are even greater than at lower-altitudes areas. This study will enrich the new understanding of vertical distribution of atmospheric components and explained the strong AOC on the TP. From the point of view of the data and scientific value, I recommend this manuscript being published in ACP after following revisions.

1. Authors should reorganize the abstract. The significance of the research and the most significant scientific findings should be fully presented.

Re: Thanks for your great comments. We have reorganized the abstract as following:

"The Tibetan Plateau (TP) plays a key role in regional environment and global climate change, however, the lack of vertical observation of atmospheric species, such as HONO and $O_3$, hinders a deeper understanding of the atmospheric chemistry and atmospheric oxidation capacity (AOC) on the TP. In this study, we conducted multi-axis differential optical absorption spectroscopy (MAX-DOAS) measurements at Nam Co, the central TP, to observe the vertical profiles of aerosol, water vapor ($H_2O$), $NO_2$, HONO and $O_3$ from May to July 2019. In addition to $NO_2$ mainly exhibiting a Gaussian shape with the maximum value appearing at 300-400 m, other four species all showed an exponential shape and decreased with the increase of height. The maximum values of monthly averaged aerosol (0.17 $km^{-1}$) and $O_3$ (66.71 ppb) occurred on May, $H_2O$ ($3.68 \times 10^{17}$ molec $cm^{-3}$) and HONO (0.13 ppb) appeared on July, while $NO_2$ (0.39 ppb) occurred on June at 200-400 m layer. $H_2O$, HONO and $O_3$ all exhibited a multi-peak pattern, and aerosol appeared a bi-peak pattern for their averaged diurnal variations. The averaged vertical profiles of OH production rates from $O_3$ and HONO all exhibited an exponential shape decreasing with the increase of height with maximum values of 2.61 ppb/h and 0.49 ppb/h at the bottom layer, respectively. The total OH production rate contributed by HONO and $O_3$ on the TP was obviously larger than that in low-altitude areas. In addition, source analysis for HONO and $O_3$ at different height layers were conducted. The heterogeneous reaction of $NO_2$ on wet surfaces was a significant source of HONO. The maximum values of HONO/$NO_2$ appeared around $H_2O$ being $1.0 \times 10^{17}$ molec $cm^{-3}$ and aerosol being lager 0.15 $km^{-1}$ under 1.0 km, and the maximum values usually accompanied with $H_2O$ being $1.0\text{-}2.0 \times 10^{17}$ molec $cm^{-3}$ and aerosol being lager 0.02 $km^{-1}$ at 1.0-2.0 km. $O_3$ was potentially sourced from south Asian

subcontinent and Himalayas through long-range transport. Our results enrich the new understanding of vertical distribution of atmospheric components and explained the strong AOC on the TP."

2.The description of the retrieval algorithm and the corresponding uncertainty analysis of vertical profiles is missing. The authors could even put them in the supplementary materials.

Re: Thanks for your great comments.

● *Vertical profile retrieval algorithm*

The atmospheric vertical profile (aerosol, $H_2O$, $NO_2$, HONO, and $O_3$) retrieval algorithm from MAX-DOAS measurements was developed based on the optimal estimation method and used radiative transfer model as the forward model (Lin et al., 2020; Liu et al., 2022; Ji et al., 2023; Xing et al., 2023). The maximum a posteriori state vector $x$ is determined by minimizing the following cost function $\chi^2$.

$$\chi^2 = \left(y - F\left(x,b\right)\right)^T S_\varepsilon^{-1} \left(y - F\left(x,b\right)\right) + \left(x - x_a\right)^T S_a^{-1} \left(x - x_a\right)$$

Here, $F\left(x,b\right)$ is the forward model, which describes the measured DSCDs $y$ as a function of the retrieval state vector $x$ (i.e., aerosol and trace gas vertical profiles) and the meteorological parameters $b$ (e.g., atmospheric pressure and temperature profiles); $x_a$ denotes the a priori vector that serves as an additional constraint; $S_\varepsilon$ and $S_a$ are the covariance matrices of $y$ and $x_a$, respectively. The retrieval of vertical profiles of aerosols and trace gases were classified into two steps. Firstly, we retrieved vertical aerosol profiles based on a series of retrieved $O_4$ DSCDs at different elevation angles. Secondly, the retrieved aerosol profiles were utilized as the input parameters to the radiative transfer model to retrieve $H_2O$, $NO_2$ and HONO profiles. Considering the strong $O_3$ absorption in the stratosphere, the retrieval of the tropospheric $O_3$ profile must remove the influence of stratospheric $O_3$. In this study, daily stratospheric $O_3$ profiles from TROPOMI measurements were included in the radiative transfer model simulation for tropospheric $O_3$ profile retrieval to account for the influence of stratospheric $O_3$ absorption on the retrieval.

● *Error analysis*

The error sources can be divided into four different types: smoothing error, noise error, forward model error, and model parameter error (Rodgers, 2004). However, in terms of this classification, some errors are difficult to be calculated or estimated. For example, the forward model error, which is caused by an imperfect representation of the physics of the system, is hard to be quantified due to the difficulty of acquiring an improved forward model. Given calculation convenience and contributing ratios of different errors in total error budget, we mainly took into account following error sources, which were smoothing and noise errors, algorithm error, cross section error, and uncertainty related to the aerosol retrieval (only for trace gas). In this study, we estimated the

contribution of different error sources to the AOD and VCDs of trace gases, and near-surface (0–200 m) trace gases' concentrations and aerosol extinction coefficients (AECs), respectively. The detailed demonstrations and estimation methods are displayed below, and the corresponding varies errors are summarized in Table R1.

a. Smoothing errors arise from the limited vertical resolution of profile retrieval. Noise errors denote the noise in the spectra (i.e., the error of DOAS fits). Considering the error of the retrieved state vector equaling the sum of these two independent errors, we calculated the sum of smoothing and noise errors on near-surface concentrations and column densities, which were 13 and 5 % for aerosols, 13 and 36 % for $H_2O$, 12 and 14 % for $NO_2$, 18 and 21 % for HONO, and 12 and 32 % for $O_3$, respectively.

b. Algorithm error is denoted by the differences between the measured and simulated DSCDs. This error contains forward model error from an imperfect approximation of forward function, parameter error of forward model, and other errors, such as detector noise (Rodgers, 2004). Algorithm error is a function of the viewing angle, and it is difficult to assign this error to each altitude. Thus, this error on the near-surface values and column densities is estimated through calculating the average relative differences between the measured and simulated DSCDs at the minimum and maximum elevation angle (except 90°), respectively (Wagner et al., 2004). In this study, we estimated these errors on the near-surface values and the column densities at 4 and 8 % for aerosols, 3 and 11 % for $NO_2$, and 20 and 20 % for HONO referring to Wang et al. (2017, 2020), 1 and 8 % for $H_2O$ referring to Lin et al. (2020), and 6 and 10 % for $O_3$ referring to Ji et al. (2023), respectively.

c. Cross section error arises from the uncertainty in the cross section. According to Thalman and Volkamer, (2013), Lin et al. (2020), Vandaele et al. (1998), Stutz et al. (2000), and Serdyuchenko et al. (2014), we adopted 4, 3, 3, 5, and 2 % for $O_4$ (aerosols), $H_2O$, $NO_2$, HONO and $O_3$, respectively.

d. The profile retrieval error for trace gases is sourced from the uncertainty of aerosol extinction profile retrieval and propagated to trace gas profile. This error could be roughly estimated based on a linear propagation of the total error budgets of the aerosol retrievals. The errors of the learned four trace gases were roughly estimated at 14 % for VCDs and 10 % for near-surface concentrations, respectively.

The total uncertainty was the sum of all above errors in the Gaussian error propagation, and the error results were listed in Table R1. We found that the smoothing and noise errors played a dominant role in the total uncertainties of aerosol and trace gases. Moreover, improving the accuracy and temperature gradient of the absorption cross section is another important means to reduce the uncertainty of the vertical profiles in the future, especially for $O_3$.

**Table R1.** Error budget estimation (in %) of the retrieved near-surface (0–200 m) concentrations of trace gases and AECs, and AOD and VCDs.

| | | Error sources | | | | Total |
|---|---|---|---|---|---|---|
| | | Smoothing and noise errors | Algorithm error | Cross section error | Related to the aerosol retrieval (only for trace gases) | |
| Near-surface | aerosol | 13 | 4 | 4 | - | 14 |
| | $H_2O$ | 13 | 1 | 3 | 14 | 19 |

| | | | | | | |
|---|---|---|---|---|---|---|
| | NO$_2$ | 12 | 3 | 3 | 14 | 18 |
| | HONO | 18 | 20 | 5 | 14 | 29 |
| | O$_3$ | 12 | 6 | 2 | 14 | 19 |
| VCD or AOD | AOD | 5 | 8 | 4 | - | 10 |
| | H$_2$O | 36 | 8 | 3 | 10 | 38 |
| | NO$_2$ | 14 | 11 | 3 | 10 | 20 |
| | HONO | 21 | 20 | 5 | 10 | 31 |
| | O$_3$ | 32 | 10 | 2 | 10 | 35 |

3.Section 2.1 emphasized HCHO, which seems to be irrelevant to this study. I suggest the authors to modificate Figure S1 to narrow the area and focus on the area around Nam Co.

Re: Thanks for your great comments. We have modified Figure S1 as following. The HCHO was removed and the study area was narrowed down.

[Figure]

Figure R2 (S1). Averaged spatial distributions of (a) AOD monitored by Himawari-8, (b) NO$_2$ VCDs monitored by TROPOMI, (and c) O$_3$ total VCDs monitored by OMI from May to July 2019 in Nam Co.

4.The authors concluded "high concentration NO$_2$ should be attributed to the transport process from the NOx produced by ice and snow on the top of Mt. Tanggula". Can you give adequate data and literature support?

Re: Thanks for your great comments. Firstly, we added two important literatures as following to support this conclusion.

(1) Fisher F N. Extinction of UV-visible radiation in wet midlatitude (maritime) snow: Implications for increased NO$_x$ emission. Journal of Geophysical Research, 110, D21301, doi:10.1029/2005JD005963, 2005. *The main point was that snow cover in mid-latitude mountainous areas contributed significantly to NO$_x$ emissions.*

(2) Lin W, Wang F, Ye C, Zhu T. Observation of strong NO$_x$ release over Qiyi Glacier, China. The Cryosphere, doi.org/10.5194/tc-2021-32, 2021. *The main point was that high NO$_x$ production was due to photochemical reactions on the snow surface of the Tibetan Plateau.*

Secondly, we also did WPSCF analysis, and the WPSCF passing through Mt. Tanggula showed high values at 300-400 m layer, especially at 400 m (> 0.3). It indicated that the important contribution to NO$_x$ from ice and snow on the top of Mt. Tanggula under strong ultraviolet radiation.

[Figure]

Figure R3 (S3). Spatial distributions of 24-h WPSCF values for $NO_2$ at (a) 300 m, and (b) 400 m height layers from 01 May to 09 July 2019 over CAS (NAMORS).

5. P6 L194-197, "Moreover, the large-scaled spatial distributions of AOD, $O_3$, $NO_2$ and HCHO over CAS (NAMORS) were monitored by Himawari-8 (Bessho et al., 2016), OMI (Veefkind et al., 2004) and TROPOMI (Griffin et al., 2018; Su et al., 2020), respectively."
Re: Thanks for your great comments. Referred to comment 3, we removed HCHO, and rewritten this sentence as "Moreover, the large-scaled spatial distributions of AOD, $O_3$ and $NO_2$ over CAS (NAMORS) were monitored by Himawari-8 (Bessho et al., 2016), OMI (Veefkind et al., 2004) and TROPOMI (Griffin et al., 2018; Su et al., 2020), respectively."

6. P6L202, increases > increased
Re: Thanks for your great comments. We have rewritten this sentence as "Subsequently, the AOD increases significantly, reaching maximum values during 15:00-17:00 (average of 0.107km-1), which was about 1.408 times the diurnal average value.".

7. P7L212-213, shown > was, and > but
Re: Thanks for your great comments. We have rewritten this sentence as "As shown in

Figure S2, the diurnal variation of PBL in Nam Co from May to July 2019 was lower in the early morning and late afternoon, but higher between 11:00 and 17:00 with the maximum PBL larger than 2.0 km.".

8. P12L390, first > firstly

Re: Thanks for your great comments. We have rewritten this sentence as "This phenomenon of HONO/NO$_2$ firstly increasing and then decreasing with the increasing of water vapor (or relative humidity) was usually found in low-altitude areas in previous studies (Wang et al., 2013; Liu et al., 2019; Xing et al., 2021; Xu et al., 2021).".

**References**

Fisher F N. Extinction of UV-visible radiation in wet midlatitude (maritime) snow: Implications for increased $NO_x$ emission. Journal of Geophysical Research, 110, D21301, doi:10.1029/2005JD005963, 2005.

Lin W, Wang F, Ye C, Zhu T. Observation of strong $NO_x$ release over Qiyi Glacier, China. The Cryosphere, doi.org/10.5194/tc-2021-32, 2021.

Ji X, Liu C, Wang Y, Hu Q, Lin H, Zhao F, Xing C, Tang G, Zhang J, Wagner T. Ozone profiles without blind area retrieved from MAX-DOAS measurements and comprehensive validation with multi-platform observations. Remote Sensing of Environment, 284, 113339, doi.org/10.1016/j.res.2022.113339, 2023.

Liu C, Xing C, Hu Q, Li Q, Liu H, Hong Q, Tan W, Ji X, Lin H, Lu C, Lin J, Liu H, Wei S, Chen J, Yang K, Wang S, Liu T, Chen Y. Ground-based hyperspectral stereoscopic remote sensing network: A promising strategy to learn coordinated control of $O_3$ and $PM_{2.5}$ over China. Engineering, 19, 71-83, doi.org/10.1016/j.eng.2021.02.019, 2022.

Lin H, Liu C, Xing C, Hu Q, Hong Q, Liu H, Li Q, Tan W, Ji X, Wang Z, Liu J. Validation of water vapor vertical distributions retrieved from MAX-DOAS over Beijing, China. Remote Sensing, 12, 3193, doi.org/10.3390/rs12193193, 2020.

Xing C, Xu S, Song Y, Liu C, Liu Y, Lu K, Tan W, Zhang C, Hu Q, Wang S, Wu H, Lin H. A new insight into the vertical differences in $NO_2$ heterogeneous reaction to produce HONO over inland and marginal seas. Atmospheric Chemistry and Physics, 23, 5815-5834, doi.org/10.5194/acp-23-5815-2023, 2023.

Rodgers C D. Inverse methods for atmospheric sounding: theory and practice. Singapore-New Jersey-London-Hong: World Scientific Publishing; 2000.

Wagner T, Dix B, FriedeBurg C V, Frieß U, Sanghavi S, Sinreich R, Platt U. MAX-DOAS $O_4$ measurements: A new technique to derive information on atmospheric aerosols-Principles and information content. Journal of Geophysical Research: Atmospheres, 109, D22205, doi.org/10.1029/2004jd004904, 2004.

Serdyuchenko A, Gorshelev V, Weber M, Chehade W, Burrows J P. High spectral resolution ozone absorption cross-sections-Part 2: Temperature dependence. Atmospheric Measurement Techniques, 7, 625-636, doi:10.5194/amt-7-625-2014, 2014.

Wang Y, Lampel J, Xie P, Beirle S, Li A, Wu D, Wagner T. Ground-based MAX-DOAS observations of tropospheric aerosols, $NO_2$, $SO_2$ and HCHO in Wuxi, China, from 2011 to 2014. Atmospheric Chemistry and Physcis, 17, 2189-2215, doi.org/10.5194/acp-17-2189-2017, 2017.

Wang Y, Apituley A, Bais A, Beirle S, Benavent N, Borovski A, Bruchkouski I, Chan K L, Donner S, Drosoglou T, Finkenzeller H, Friedrich M M, Frieß U, Garcia-Nieto D, Gómez-Martín L, Hendrick F, Hilboll A, Jin J, Johnston P, Koenig T K, Kreher K, Kumar V, Kyuberis A, Lampel J, Liu C, Liu H, Ma J, Polyansky O L, Postylyakov O, Querel R, Saiz-Lopez A, Schmitt S, Tian X, Tirpitz J L, Van Roozendeal M, Volkamer R, Wang Z, Xie P, Xing C, Xu J, Yela M, Zhang C, Wagner T. Inter-comparison of MAX-DOAS measurements of tropospheric HONO slant column densities and vertical

profiles during the CINDI-2 campaign. Atmospheric Measurement Techniques, 13, 5087–5116, doi.org/10.5194/amt-13-5087-2020, 2020.

Thalman R, Volkamer R. Temperature dependent absorption cross-sections of $O_2$-$O_2$ collision pairs between 340 and 630 nm and at atmospherically relevant pressure. Physical Chemistry Chemical Physics, 15, 15371-15381, 2013.

Vandaele A C, Hermans C, Simon P C, Carleer M, Colin R, Fally S, Merienne M F, Jenouvrier A, Coquart D. Measurements of the $NO_2$ absorption cross-section from 42000 cm$^{-1}$ to 10000 cm$^{-1}$ (238–1000nm) at 220K and 294K. Journal of Quantitative Spectroscopy and Radiative Transfer, 59, 171-184, 1998.

Stutz J, Kim E S, Platt U, Bruno P, Perrino C, Febo A. UV-visible absorption cross sections of nitrous acid. Journal of Geophysical Research: Atmospheres, 105, 14585-14592, 2000.

---

## Author Comment (AC2)

Point-to-point responses

*We appreciate the reviewers for their valuable and constructive comments, which are very helpful for the improvement of the manuscript. We have revised the manuscript carefully according to the reviewers' comments. We have addressed the reviewers' comments on a point-to-point basis as below for consideration, where the reviewers' comments are cited in **black**, and the responses are in **blue**.*

**Referee #2**

This manuscript describes ground-based MAX-DOAS measurements in Nam Co over the Tibetan Plateau (TP). The measurements are used to obtain vertical distributions of several atmospheric components (aerosol, $H_2O$, $NO_2$, HONO and $O_3$) via spectral analysis and OEM profile retrieval. The data are further used to analyze the temporal and vertical evolutions for these species. The OH production as well as possible daytime HONO and $O_3$ sources were also discussed during the field campaign. Overall, observing and investigating the vertical profiles of atmospheric components at the background station in the relatively remote and high-altitude region are significant. However, two concerns at least should be clarified in detail before the manuscript is considered to be accepted.

(1) Firstly, the reliabilities of vertical profile of atmospheric components (aerosol, $H_2O$, $NO_2$, HONO and $O_3$) should be validated. Just using the parameter setting scheme of spectral analysis and profile retrieval in previous studies won't do for the specific station over the TP. For example, the HONO spectral structures are almost drowned in the noise in Figure 2d. The sensitivity tests about the parameter setting scheme of spectral analysis and profile retrieval should be presented in detail. In addition, just according to the comparison of surface multi-source data in Section 3.3, it cannot illustrate the reliabilities of vertical profiles. Also, the correlation was weak for HONO ($R^2$=0.38) with larger deviations in Figure 6b.

Re: Thanks for your great comments.

We added a section of "*2.2.3 Error analysis*" to ensure the reliability of data.

The error sources can be divided into four different types: smoothing error, noise error, forward model error, and model parameter error (Rodgers, 2004). However, in terms of this classification, some errors are difficult to be calculated or estimated. For example, the forward model error, which is caused by an imperfect representation of the physics of the system, is hard to be quantified due to the difficulty of acquiring an improved forward model. Given calculation convenience and contributing ratios of different errors in total error budget, we mainly took into account following error sources, which were smoothing and noise errors, algorithm error, cross section error, and uncertainty related to the aerosol retrieval (only for trace gas). In this study, we estimated the contribution of different error sources to the AOD and VCDs of trace gases, and near-surface (0–200 m) trace gases' concentrations and aerosol extinction coefficients

(AECs), respectively. The detailed demonstrations and estimation methods are displayed below, and the corresponding varies errors are summarized in Table R1.

a. Smoothing errors arise from the limited vertical resolution of profile retrieval. Noise errors denote the noise in the spectra (i.e., the error of DOAS fits). Considering the error of the retrieved state vector equaling the sum of these two independent errors, we calculated the sum of smoothing and noise errors on near-surface concentrations and column densities, which were 13 and 5 % for aerosols, 13 and 36 % for $H_2O$, 12 and 14 % for $NO_2$, 18 and 21 % for HONO, and 12 and 32 % for $O_3$, respectively.

b. Algorithm error is denoted by the differences between the measured and simulated DSCDs. This error contains forward model error from an imperfect approximation of forward function, parameter error of forward model, and other errors, such as detector noise (Rodgers, 2004). Algorithm error is a function of the viewing angle, and it is difficult to assign this error to each altitude. Thus, this error on the near-surface values and column densities is estimated through calculating the average relative differences between the measured and simulated DSCDs at the minimum and maximum elevation angle (except 90°), respectively (Wagner et al., 2004). In this study, we estimated these errors on the near-surface values and the column densities at 4 and 8 % for aerosols, 3 and 11 % for $NO_2$, and 20 and 20 % for HONO referring to Wang et al. (2017, 2020), 1 and 8 % for $H_2O$ referring to Lin et al. (2020), and 6 and 10 % for $O_3$ referring to Ji et al. (2023), respectively.

c. Cross section error arises from the uncertainty in the cross section. According to Thalman and Volkamer, (2013), Lin et al. (2020), Vandaele et al. (1998), Stutz et al. (2000), and Serdyuchenko et al. (2014), we adopted 4, 3, 3, 5, and 2 % for $O_4$ (aerosols), $H_2O$, $NO_2$, HONO and $O_3$, respectively.

d. The profile retrieval error for trace gases is sourced from the uncertainty of aerosol extinction profile retrieval and propagated to trace gas profile. This error could be roughly estimated based on a linear propagation of the total error budgets of the aerosol retrievals. The errors of the learned four trace gases were roughly estimated at 14 % for VCDs and 10 % for near-surface concentrations, respectively.

The total uncertainty was the sum of all above errors in the Gaussian error propagation, and the error results were listed in Table R1. We found that the smoothing and noise errors played a dominant role in the total uncertainties of aerosol and trace gases. Moreover, improving the accuracy and temperature gradient of the absorption cross section is another important means to reduce the uncertainty of the vertical profiles in the future, especially for $O_3$.

**Table R1.** Error budget estimation (in %) of the retrieved near-surface (0–200 m) concentrations of trace gases and AECs, and AOD and VCDs.

| | | Error sources | | | | Total |
|---|---|---|---|---|---|---|
| | | Smoothing and noise errors | Algorithm error | Cross section error | Related to the aerosol retrieval (only for trace gases) | |
| Near-surface | aerosol | 13 | 4 | 4 | - | 14 |
| | $H_2O$ | 13 | 1 | 3 | 14 | 19 |
| | $NO_2$ | 12 | 3 | 3 | 14 | 18 |
| | HONO | 18 | 20 | 5 | 14 | 29 |

| | | | | | | |
|---|---|---|---|---|---|---|
| | O₃ | 12 | 6 | 2 | 14 | 19 |
| VCD or AOD | AOD | 5 | 8 | 4 | - | 10 |
| | H₂O | 36 | 8 | 3 | 10 | 38 |
| | NO₂ | 14 | 11 | 3 | 10 | 20 |
| | HONO | 21 | 20 | 5 | 10 | 31 |
| | O₃ | 32 | 10 | 2 | 10 | 35 |

Certainly, we also did independent validations. However, there was not other vertical observations during our campaign on the TP. Therefore, we did validations between in situ measurements and the bottom layer of MAX-DOAS profiles (Figure R1).

[Figure]

Figure R1. Validations of (a) MAX-DOAS NO₂ vs in situ NO₂, (b) MAX-DOAS HONO vs LOPAP HONO, (c) MAX-DOAS O₃ vs in situ O₃.

Moreover, we did vertical-profile validations in Shanghai and Beijing. As shown in Figure R2, we used Mie lidar to validate MAX-DOAS aerosol vertical profiles, and used balloon-based NO₂ profiles to validate MAX-DOAS NO₂ vertical profiles (Xing et al., 2017). The good agreement indicates the reliability of MAX-DOAS retrieved aerosol and NO₂ profiles.

[Figure]

Figure R2. Validations of (a) MAX-DOAS aerosol profile vs lidar aerosol profile, (b) MAX-DOAS NO₂ profile vs balloon-based NO₂ profile.

As shown in Figure R3, the retrieved O₃ profiles were validated with tower-based O₃ profiles in Beijing. The correlation coefficients (R) between them on 2 m, 60 m, 160 m, and 280 m were 0.844, 0.864, 0.883 and 0.902, respectively.

[Figure]

Figure R3. Validations of MAX-DOAS $O_3$ profile vs tower-based $O_3$ profile at four height layers (a) 2 m, (b) 60 m, (c) 160 m, and (d) 280 m.

As shown in Figure R4, we re-selected a representative HONO fitting result. We can be sure that the HONO results used in the manuscript all from well spectra fittings. Considering the effects of low concentration and weak absorption of HONO, and spectral noise, we performed third-order averaging of the spectra during spectral analysis.

[Figure]

Figure R4. DOAS fit examples of $O_4$, $H_2O$, $NO_2$, HCHO, tropospheric $O_3$ and stratospheric $O_3$. The red line and black line represent the measured and fitted results, respectively.

About the weak correlation of HONO (R=0.62) between MAX-DOAS measurements and LOPAP measurements, we make it clear that uncertainties also appear in the LOPAP HONO results during the observation period (Wang et al., 2023). Longer HONO observations on the TP based on more instruments are needed in the future, in order to further analyze the uncertainty of HONO observations among different instruments.

(2) The manuscript tried to analyze the OH production (Section 4.1) from HONO and $O_3$ at different height layers through vertical observations and TUV calculations. The TUV model is suitable for exploring the photolysis rates, but OH production is determined by complex chemical process involving many atmospheric components. The equations in Line 342 and 343 are simplified for studying OH production. It should present more details on the parameter scheme during simulation by TUV model, and more comparisons of OH production simulated by other models if possible.

Re: Thanks for your great comments.

In this study, we used online TUV model.

*https://www2.acom.ucar.edu/modeling/tropospheric-ultraviolet-and-visible-tuv-radiation-model*

The main parameters for this model were cloud information, total ozone column, aerosol optical depth (AOD), single scattering albedo (SSA), and Ångström exponents. In this study, we selected clean and cloud free days (Figure R6), the AOD at 361 nm was derived from aerosol extinction profiles measured by MAX-DOAS; the daily total ozone column density was measured by TROPOMI with a value range of 260-280 DU; the single scattering albedo (SSA) was calculated based on the regression analysis of multi-wavelength (361 and 477 nm) $O_4$ absorptions measured by MAX-DOAS (Xing et al., 2019); fixed Ångström exponents of 0.508, 0.581 and 0.713 were used in May, June and July, respectively, referring to Xia et al. (2011).

The parameter scheme and the corresponding code of TUV model can be found at:

*https://www2.acom.ucar.edu/modeling/tuv-download*

***We also referred to He et al. (2023) to calculate the photolysis rate as following:***

■ Photolysis rate

The photolysis rate of a given molecule $X$, $J_X$, is dependent on the incident actinic flux, $F(\lambda)$, the absorption cross-section, $\sigma(\lambda)$, and the quantum yield, $\phi(\lambda)$:

$$J_X = \int_{\lambda_1}^{\lambda_2} \sigma(\lambda)\phi(\lambda)F(\lambda)d\lambda$$

$\sigma(\lambda)$ and $\phi(\lambda)$ (both of the two parameters depend on wavelength and temperature) were retrieved from the MPI-Mainz UV/VIS Spectral Atlas (http://satellite.mpic.de/spectral_atlas). The actinic flux $F(\lambda)$ describes the total

energy pf photons incident on the unit sphere, which is calculated by integrating the radiance over all directions and can be expressed as:

$$F(\lambda) = \int_0^{2\pi} \int_{-\pi/2}^{\pi/2} L(\lambda, \theta, \varphi) \sin \theta d\theta d\varphi$$

$L(\lambda, \theta, \varphi)$ is spectral radiance ( $\theta$ and $\varphi$ are the zenith and azimuth angles, respectively).

Theoretical relationships between the actinic flux and the irradiance were presented to convert actinic flux from irradiance:

$$F(\lambda) = \left( \alpha(\lambda) + \left( \frac{1}{\cos(\theta)} - \alpha(\lambda) \right) \frac{E_{dir}(\lambda)}{E(\lambda)} \right) \times E(\lambda)$$

$\alpha(\lambda)$ is the spectral ratio of diffuse actinic flux to diffuse irradiance. $E_{dir}(\lambda)/E(\lambda)$ is the spectral ratio of direct to total irradiance. $\theta$ is the solar zenith angle. By definition, the spectral irradiance $E(\lambda)$ is the radiation incident on unit flat (usually horizontal) surface, which is calculated by integration of the radiance over a hemisphere and can be expressed as:

$$E(\lambda) = \int_0^{2\pi} \int_0^{\pi/2} L(\lambda, \theta, \varphi) \cos \theta \sin \theta d\theta d\varphi$$

The photolysis rate and actinic flux at different altitudes (h) can be described as:

$$F(h, \lambda) = \left( \alpha(h, \lambda) + \left( \frac{1}{\cos(\theta)} - \alpha(h, \lambda) \right) \frac{E_{dir}(\lambda)}{E(\lambda)} \right) \times E(h, \lambda)$$

■ Spectral irradiance at different altitudes

In the absence of actinic flux observation, it seems a promising method that the photolysis rate with acceptable uncertainty is obtained by calculating the actinic flux from the spectral irradiance. The solar spectral irradiance at different altitudes can be calculated as following:

$$E(h, \lambda) = E_{extra}(\lambda) \cdot e^{-\frac{C(\lambda) \cdot D(h)}{\cos \theta}}$$

$E_{extra}(\lambda)$ is solar spectral irradiance from extraterrestrial. $C(\lambda)$ represents the effect of wavelength on absorption coefficient (depend on wavelength and weather conditions). $D(h)$ reflects the influence of altitude on absorption coefficient and is defined as the ratio of atmospheric pressure at $h$ km ($P_h$) to that at 0.0 km ($P_0$), i.e. $D(h) = P_h/P_0$. By conversion, $E(h, \lambda)$ can be calculated as following:

$$\ln E(h, \lambda) = \left( \ln E(0, \lambda) - \ln E_{extra}(\lambda) \right) \times \frac{P_h}{P_0} + \ln E_{extra}(\lambda)$$

$P_0$ came from ERA5, while $P_h$ can be converted from height by barometric height formula. Solar spectral irradiance at ground surface $E(0, \lambda)$ can be calculated using

total surface solar radiation and MAX-DOAS measured spectra.

- $\alpha(h,\lambda)$ and $E_{dir}(h,\lambda)/E(h,\lambda)$ at different altitudes

$\alpha(h,\lambda)$ is the spectral ratio of diffuse actinic flux to diffuse irradiance at different altitudes, which were affected by different factors (e.g. solar zenith angle and wavelength). Considering the almost constant surface albedo during the campaign, the impacts of solar zenith angle on the change of $\alpha(0,\lambda)$ is small and could be ignored. The effect of wavelength can be estimated as following equation. As discussed previously, due to lack of studies on $\alpha(0,\lambda)$ at different altitudes, we have to apply it to higher altitudes in this study.

$$\alpha(h,\lambda)=\frac{1}{0.5-(\lambda-300)/1000}$$

$E_{dir}(h,\lambda)/E(h,\lambda)$ is the spectral ratio of direct irradiance to total irradiance at different altitudes. The error of $E_{dir}(h,\lambda)/E(h,\lambda)$ has relatively less influence on the photolysis rate, especially during the midday with low solar zenith angle. Since the solar radiation can be defined as the total energy of the full spectral irradiance integral, we assume that $E_{dir}(h,\lambda)/E(h,\lambda)$ and $G_{dir}(h)/G(h)$ were approximately equal in this research, and further can be quantified by following equation according to the national guidelines (GB/T 37525-2019).

$$\frac{E_{dir}(h,\lambda)}{E(h,\lambda)}=\frac{G_{dir}(h)}{G(h)}=1-f(k,h)$$

$G_{dir}(h)$ is direct solar radiation ( $G_{dir}(h)=\int_0^{3000} E_{dir}(h,\lambda)d\lambda$ ). $G(h)$ is total solar radiation ( $G(h)=\int_0^{3000} E(h,\lambda)d\lambda$ ). $f(k,h)$ represents the ratio of diffuse solar radiation to total solar radiation.

$$f(k,h)=\begin{cases}1.0-0.249\cdot k(h) & 0\le k(h)\le 0.35\\ 1.557-1.84\cdot k(h) & 0.35\le k(h)\le 0.75\\ 0.177 & k(h)\ge 0.75\end{cases}$$

$$k(h)=G(h)/G_{extra}$$

where $k(h)$ is clearness coefficient. $n$ is day of year (DOY) and $S_0$ is the solar constant of 1366.1 W·m$^{-2}$; $G_{extra}$ is extraterrestrial solar radiation

$(G_{extra} = S_0 \cdot (1 + 0.033\cos\left(\dfrac{360n}{365}\right)) \cdot \cos\theta)$.

Total solar radiation at differently altitudes *G(h)* can be estimated by the following equation.

$$G(h) = G_{extra} \cdot e^{-\dfrac{C \cdot D(h)}{\cos\theta}}$$

where *C* represents absorption coefficient (Depend on weather conditions). With conversion, it can be calculated by following equation.

$$\ln G(h) = \left(\ln G(0) - \ln G_{extra}\right) \times \dfrac{P_h}{P_0} + \ln G_{extra}$$

The *G(0)* can be observed by meteorological instruments or reanalysis data.

The correlation coefficients (R) between J(HONO) calculated using above method and J(HONO) simulated by TUV model at different altitude were **larger than 0.91**. It indicates that the HONO photolysis rates simulated by the TUV model are accurate under clean and cloud-free days. We also plan to optimize the parameterization scheme of WRF-Chem to simulate OH in the future.

In addition, the calculated OH were validated with measured OH, but the specific discussion will be organized in a separate study.

**Specific comments**

1. Authors' affiliations and addresses should be numbered in the appearance order of their names, to avoid the misunderstanding of academic misconduct.

Re: Thanks for your great comments. We have modified this section as following:

[revised manuscript text omitted]

(17) The initial input parameters were as follows: the AOD at 361 nm was derived from aerosol extinction profiles measured by MAX-DOAS.

(18) The PBL height was simulated using WRF with spatiotemporal resolutions of 20✗20 $km^2$ and 1.0 hour (detailed configurations in Sect. S2 of the supplement).

[revised manuscript text omitted]

4. Line 26-38: This paragraph is not closely related to the key ideas of this paper. Moreover, the last sentence is lack of proper deduction.

Re: Thanks for your great comments. We try to make some explanations and modifications as following:

(1) The purpose of this paragraph is to emphasize the importance of the TP on climate change and reginal air pollution due to its special location and geographic features. It amounts to an explanation of the context of this study.

(2) We have rewritten this paragraph as "The TP spans 2.5 million square kilometers with an average altitude of over 4000 m. Therefore, the TP is called the "Third Pole" of the earth (Ma et al., 2020; Kang et al., 2022). It is the home to tens of thousands of glaciers and nourishes more than 10 of Asia's rivers, thus it also acts the role of "Water Tower of Asia" (Qu et al., 2019; Ma et al., 2022). Due to its special topography, the TP is the heat source of atmosphere due the strong solar radiation, which as the driven force to profoundly affect the regional atmospheric circulation, global weather conditions and climate change (Yanai et al., 1992; Boos et al., 2010; Chen et al., 2015; Liu et al., 2022; Zhou et al., 2022). Monsoon rainfall in Asia, flood over the Yangtze River valley, and El Niño in the Pacific Ocean are strongly associated with the TP (Hsu et al., 2003; Li et al., 2016; Lei et al., 2019). In addition, the cyclone circulations caused by the TP heat source also can inhibit the diffusion of atmospheric pollutants in the areas around the TP, such as the Sichuan Basin, causing regional pollution (Zhang et al., 2019). Therefore, observations of the atmospheric species on the TP are essential to enhance the in-depth understanding of its atmospheric physicochemical processes."

5. Line 39-73: This paragraph is lack of many key references for MAX-DOAS observations over the Tibetan Plateau. It is not appropriate to cite only the studies from author's group.

Re: Thanks for your great comments. However, the reality is that there are very few reports on the atmospheric environment of the Tibetan Plateau based on MAX-DOAS. These studies all have been carried out by Chinese researchers. The reference list is as following:

[1] Cheng, S., Pu, G., Ma, J., Hong, H., Du, J., Yudron, T., Wagner, T.: Retrieval of tropospheric $NO_2$ vertical column densities from ground-based MAX-DOAS measurements in Lhasa, a city on the Tibetan Plateau, Remote Sens., 15, 4689, 2023.
[2] Cheng, S., Ma, J., Zheng, A., Gu, M., Donner, S., Donner, S., Zhang, W., Du, J., Li, X., Liang, Z., Lv, J., Wagner, T.: Retrieval of $O_3$, $NO_2$, BrO and OClO columns from ground-based zenith scattered light DOAS measurements in summer and autumn over

the Northern Tibetan Plateau, Remote Sens., 13, 4242, 2021.

[3] Ma, J., Donner, S., Donner, S., Jin, J., Cheng, S., Guo, J., Zhang, Z., Wang, J., Liu, P., Zhang, G., Pukite, J., Lampel, J., Wagner, T.: MAX-DOAS measurements of $NO_2$, $SO_2$, HCHO, and BrO at the Mt. Waliguan WMO GAW global baseline station in the Tibetan Plateau, Atmos. Chem. Phys., 20, 6973-6990, 2020.

[4] Cheng, S., Cheng, X., Ma, J., Xu, X., Zhang, W., Lv, J., Bai, G., Chen, B., Ma, S., Ziegler, S., Donner, S., Wagner, T.: Mobile MAX-DOAS observations of tropospheric $NO_2$ and HCHO during summer over the Three Rivers' Source region in China, Atmos. Chem. Phys., 23, 3655-3677, 2023.

[5] Xing, C., Liu, C., Wu, H., Lin, J., Wang, F., Wang, S., Gao, M.: Ground-based vertical profile observations of atmospheric composition on the Tibetan Plateau (2017-2019), Earth Syst. Sci. Data, 13, 4897-4912, 2021.

[6] Li, M., Mao, J., Chen, S., Bian, J., Bai, Z., Wang, X., Chen, W., Yu, P.: Significant contribution of lightning $NO_x$ to summertime surface $O_3$ on the Tibetan Plateau, Sci. Total Environ., 829, 154639, 2022.

Moreover, we have rewritten the related sentence as "MAX-DOAS has the technical advantage of low-cost continuous observation of multiple atmospheric components (i.e. aerosol, $O_3$ and their precursors) (Wang et al., 2018; Ma et al., 2020; Cheng et al., 2021; Xing et al., 2021; Li et al., 2022; Cheng et al., 2023a, 2023b).".

6. Line 75: The $O_3$ will be diluted when the air mass comes from clean source regions, such as marine atmosphere.

Re: Thanks for your great comments. We would like to make two statements about this problem as following:

The main point referred from Ye and Gao (1997) is that the strong convergent airflow formed under the combined action of monsoon, subtropical anticyclone and the airflow of subtropical westerlies could promote the accumulation of $O_3$ on the TP in summer. It is not that air mass coming from clean regions promote the increase of $O_3$ on the TP, but rather that the strong convergent airflow contributed by monsoon, subtropical anticyclone and the airflow of subtropical westerlies could promote the accumulation of $O_3$ on the TP. The strong convergent airflow not only promote the transport of high-concentration $O_3$ from the surrounding area to the TP but also inhibit the diffusion of $O_3$ from the TP to the outside.

Moreover, we retrieved the spatial distribution of tropospheric $O_3$ based on the issues raised by the reviewers (Figure R5). We could find that there are some areas with high $O_3$ values in the oceans, especially in the Indian Ocean and in the Bohai and Yellow Sea regions of China. The sources of $O_3$ with high values in these regions need to be further explored.

*Reference:* Ye, D. Z., and Gao, Y. X.: The meteorology of the Tibetan Plateau (in Chinese), 278pp., Science Press, Beijing, pp. 39-48, 1979.

[Figure]

Figure R5. Spatial distributions of tropospheric $O_3$ monitored by TROPOMI.

7. Line 122: Again, sensitivity test and profile validation by independent data should be presented to confirm the reliability of retrieval results.
Re: Thanks for your great comments. We have reorganized this section as following:
*2 Method and methodology*
*2.1 Site*
*2.2 Measurements*
*2.2.1 Instrument setup and spectral analysis*
*2.2.2 Vertical profile retrieval*
*2.2.3 Error analysis*
In order to ensure the reliability of data, error analysis was carried out.
The error sources can be divided into four different types: smoothing error, noise error, forward model error, and model parameter error (Rodgers, 2004). However, in terms of this classification, some errors are difficult to be calculated or estimated. For example, the forward model error, which is caused by an imperfect representation of the physics of the system, is hard to be quantified due to the difficulty of acquiring an improved forward model. Given calculation convenience and contributing ratios of different errors in total error budget, we mainly took into account following error sources, which were smoothing and noise errors, algorithm error, cross section error, and uncertainty related to the aerosol retrieval (only for trace gas). In this study, we estimated the contribution of different error sources to the AOD and VCDs of trace gases, and near-surface (0–200 m) trace gases' concentrations and aerosol extinction coefficients (AECs), respectively. The detailed demonstrations and estimation methods are displayed below, and the corresponding varies errors are summarized in Table R1.

e. Smoothing errors arise from the limited vertical resolution of profile retrieval. Noise errors denote the noise in the spectra (i.e., the error of DOAS fits). Considering the error of the retrieved state vector equaling the sum of these two independent errors, we calculated the sum of smoothing and noise errors on near-surface concentrations and column densities, which were 13 and 5 % for aerosols, 13 and 36 % for $H_2O$, 12 and 14 % for $NO_2$, 18 and 21 % for HONO, and 12 and 32 % for $O_3$, respectively.

f. Algorithm error is denoted by the differences between the measured and simulated DSCDs. This error contains forward model error from an imperfect approximation

of forward function, parameter error of forward model, and other errors, such as detector noise (Rodgers, 2004). Algorithm error is a function of the viewing angle, and it is difficult to assign this error to each altitude. Thus, this error on the near-surface values and column densities is estimated through calculating the average relative differences between the measured and simulated DSCDs at the minimum and maximum elevation angle (except 90°), respectively (Wagner et al., 2004). In this study, we estimated these errors on the near-surface values and the column densities at 4 and 8 % for aerosols, 3 and 11 % for $NO_2$, and 20 and 20 % for HONO referring to Wang et al. (2017, 2020), 1 and 8 % for $H_2O$ referring to Lin et al. (2020), and 6 and 10 % for $O_3$ referring to Ji et al. (2023), respectively.

g. Cross section error arises from the uncertainty in the cross section. According to Thalman and Volkamer, (2013), Lin et al. (2020), Vandaele et al. (1998), Stutz et al. (2000), and Serdyuchenko et al. (2014), we adopted 4, 3, 3, 5, and 2 % for $O_4$ (aerosols), $H_2O$, $NO_2$, HONO and $O_3$, respectively.

h. The profile retrieval error for trace gases is sourced from the uncertainty of aerosol extinction profile retrieval and propagated to trace gas profile. This error could be roughly estimated based on a linear propagation of the total error budgets of the aerosol retrievals. The errors of the learned four trace gases were roughly estimated at 14 % for VCDs and 10 % for near-surface concentrations, respectively.

The total uncertainty was the sum of all above errors in the Gaussian error propagation, and the error results were listed in Table R1. We found that the smoothing and noise errors played a dominant role in the total uncertainties of aerosol and trace gases. Moreover, improving the accuracy and temperature gradient of the absorption cross section is another important means to reduce the uncertainty of the vertical profiles in the future, especially for $O_3$.

Certainly, we also did independent validations. However, there was not other vertical observations during our campaign on the TP. Therefore, we did validations between in situ measurements and the bottom layer of MAX-DOAS profiles (Figure R1).

Moreover, we did vertical-profile validations in Shanghai and Beijing. As shown in Figure R2, we used Mie lidar to validate MAX-DOAS aerosol vertical profiles, and used balloon-based $NO_2$ profiles to validate MAX-DOAS $NO_2$ vertical profiles (Xing et al., 2017). The good agreement indicates the reliability of MAX-DOAS retrieved aerosol and $NO_2$ profiles.

As shown in Figure R3, the retrieved $O_3$ profiles were validated with tower-based $O_3$ profiles in Beijing. The correlation coefficients (R) between them on 2 m, 60 m, 160 m, and 280 m were 0.844, 0.864, 0.883 and 0.902, respectively.

8. Line 156: The cross sections are lack of references in Table 1.

Re: Thanks for your great comments. We have added the corresponding references as following.

[1] Aliwell, S. R., Van Roozendael, M., Johnston, P. V., Richter, A., Wagner, T., Arlander, D. W., Burrows, J. P., Fish, D. J., Jones, R. L., Tørnkvist, K. K., Lambert, J. C., Pfeilsticker, K., and Pundt, I.: Analysis for BrO in zenith-sky spectra: an intercomparison exercise for analysis improvement, J. Geophys. Res., 107, ACH 10-1–

ACH 10-20, https://doi.org/10.1029/2001JD000329, 2002.

[2] Vandaele, A. C., Hermans, C., Simon, P. C., Carleer, M., Colin, R., Fally, S., Mérienne, M. F., Jenouvrier, A., and Coquart, B.: Measurements of the $NO_2$ absorption cross section from 42000 $cm^{-1}$ to 10000 $cm^{-1}$ (238–1000nm) at 220 K and 294 K, J. Quant. Spectrosc. Ra., 59, 171–184, 1998.

[3] Meller, R. and Moortgat, G. K.: Temperature dependence of the absorption cross sections of formaldehyde between 223 and 323 K in the wavelength range 225–375 nm, J. Geophys. Res., 105, 7089–7101, 2000.

[4] Volkamer, R., Spietz, P., Burrows, J., Platt, U.: High-resolution absorption cross-section of glyoxal in the UV-vis and IR spectral ranges, J. Photochem. Photobiol. A Chem., 172, 35–46, 2005.

[5] Rothman, L. S., Gordon, I. E., Barbe, A., Benner, D. C., Bernath, P. E., Birk, M., Boudon, V., Brown, L. R., Campargue, A., Champion, J. P., Chance, K., Coudert, L. H., Dana, V., Devi, V. M., Fally, S., Flaud, J. M., Gamache, R. R., Goldman, A., Jacquemart, D., Kleiner, I., Lacome, N., Lafferty, W. J., Mandin, J. Y., Massie, S. T., Mikhailenko, S. N., Miller, C. E., Moazzen-Ahmadi, N., Naumenko, O. V., Nikitin, A. V., Orphal, J., Perevalov, V. I., Perrin, A., Predoi-Cross, A., Rinsland, C. P., Rotger, M., Simeckova, M., Smith, M. A. H., Sung, K., Tashkun, S. A., Tennyson, J., Toth, R. A., Vandaele, A. C., Vander Auwera, J.: The HITRAN 2008 molecular spectroscopic database, J. Quant. Spectrosc. Radiat. Transf., 110, 533–572, 2009.

[6] Fleischmann, O. C., Hartmann, M., Burrows, J. P., and Orphal, J.: New ultraviolet absorption cross-sections of BrO at atmospheric temperatures measured by time-windowing Fourier transform spectroscopy, J. Photoch. Photobio. A, 168, 117–132, 2004.

9. Line 165: Are only the HONO and $O_3$ data in clear sky condition used to calculate the photolysis rates. If so, there may be many missing data in summer due to clouds over the TP. This information should be explicitly mentioned in section 4.1.

Re: Thanks for your great comments.

The cloud parameters are important and complex factors influencing the photolysis rates of HONO and $O_3$. Inaccurate evaluation of cloud parameters will increase the uncertainty of above photolysis rates. In order to remove the misunderstanding that this uncertainty introduces into the assessment of AOC by HONO and $O_3$, we judged the cloud coverage firstly during our measurements. As shown in Figure R6 (a), we develop a cloud classification method based on the diurnal variations of Color Index (CI=$I_{330}/I_{360}$). To ensure a sufficient amount of valid data, we chose sunny and cloudless days to analyze the photolysis rates of HONO and $O_3$. Only the cloud coverage scenario of *II* in Fig. R6 (b) was masked. In this study, the validity of the data was more than 70%.

[Figure]

Figure R6. (a) Color Index cloud classification algorithm process. (b) Example of cloud classification.

The information was mentioned in section 2.3 (TUV model), but we have enriched the relevant information in the revised manuscript.

"In order to ensure the accuracy of model running, we only selected data in sunny and cloudless days. Moreover, we developed a cloud classification method based on the diurnal variations of Color Index ($CI=I_{330}/I_{360}$) in Figure S2."

10. Line 177: Please add references.

Re: Thanks for your great comments. We have rewritten this sentence as following:

"Moreover, the calculated backward trajectories were clustered into three groups using Ward's variance method and Angle Distance algorithm (Ward 1963; Wang et al., 2006)."

Re: Thanks for your great comments.

Figure R8 showed the diurnal variation of PBL. We could find that the maximum values of PBL (~2.0 km) at our measurement site appeared at 11:00-17:00, a relatively long

period. Considering that the pollutants were mainly distributed within the PBL, five heights under 2.0 km were selected to analyze the variations of several pollutants (aerosol, $H_2O$, $NO_2$, HONO and $O_3$) throughout the observation period.

There may be a misunderstanding. The purpose of Figure 4 is to show the variations of pollutants within the PBL during the observation period. ***We used the daily mean values of the different pollutants.*** In addition, the average diurnal variation of different pollutants was depicted in Figure 5.

[Figure]

Figure R8. The diurnal variation of PBL in Nam Co from May to July 2019. The top and bottom of the box represented 75[th] and 25[th] percentiles, respectively. The lines and dots within the boxes were the median and mean, respectively.

We have rewritten these sentences as following:

"As shown in Figure S4, the diurnal variation of PBL in Nam Co from May to July 2019 was lower in the early morning and late afternoon, but higher between 11:00 and 17:00, a relatively long period, with the maximum PBL larger than 2.0 km. Zhang et al. (2017) and Yang et al., (2017) also reported that the PBL in Nam Co was usually larger than 1.0 km during daytime in spring and summer. In order to investigate the height-dependent variations of aerosol, $H_2O$, $NO_2$, HONO and $O_3$ within the PBL during the measurements, five height layers under the PBL (0.0-0.2 km, 0.4-0.6 km, 0.8-1.0 km, 1.2-1.4 km and 1.6-1.8 km) were thus selected."

"Figure 4. Time series of daily averaged (a) aerosol extinction, (b) $H_2O$, (c) $NO_2$, (d) HONO, and (e) $O_3$ monitored by MAX-DOAS at 0-0.2, 0.4-0.6, 0.8-1.0, 1.2-1.4 and 1.6-1.8 km five height layers from 01 May to 09 July 2019."

13. Line 224-225, 228-229: The explanations are lack of proper deduction.

Re: Thanks for your great comments.

(1) *"That indicated that the aerosol was usually local-emitted at the surface, and the occasionally appearance of strong aerosol extinction at 0.4-0.6 km, such as 13[th] and 30[th] June, was associated with long-range transport from south Asia."*

[Figure]

Figure R9. Spatial distribution of WPSCF values for aerosol at 400-600 m layer on 13[th] and 30[th] June.

In order to determine the potential source locations of aerosol at 400-600 m layer, the WPSCF model was used. As shown in Figure R9, the maximum values of WPSCF for aerosol at 400-600 m layer (> 0.45) was mainly from Nepal and the north of India.

(2) *"The average concentration of $H_2O$ at 0.0-0.2 km was $2.35 \times 10^{17}$ molec cm$^{-3}$, and the ratios of $H_2O$ at 0.4-0.6 km, 0.8-1.0 km, 1.2-1.4 km and 1.6-1.8 km to those at 0.0-0.2 km were 83.40%, 68.08%, 50.64% and 35.74%, respectively, which should attribute to the transport of $H_2O$ from Indian Ocean during the monsoon and the elevated evaporation from Nam Co lake to lead to its not obvious vertical gradient."*

[Figure]

Figure R10. Spatial distributions of WPSCF values for H2O at (a) 200 m, (b) 600 m, (c) 1000 m, (d) 1400 m, and (e) 1800 m height layers from 01[st] May to 09[th] July 2019 over CAS (NAMORS).

In order to determine the potential source locations of $H_2O$ under PBL, the WPSCF model was used. As shown in Figure R10, we could find that $H_2O$ was mainly from south Asia, especially above 1000 m, driven by the Indian Ocean monsoon.

14. Line 233-234: As a short-life atmospheric component, can the $NO_x$ around Mt. Tanggula transport to the observation site? Does the transport effect affect the $NO_2$ at the bottom layer? The causes of elevated $NO_2$ layer are probably complex.

Re: Thanks for your great comments.

$NO_2$ mainly sourced from primary emission and reginal transport in the atmosphere. The primary sources of $NO_2$ include traffic, industrial and residential emissions. However, as shown in Figure R11, there are few primary emission sources around our measurement site. The largest potential area with high $NO_2$ concentrations is from Dangxiong country, but there is ~34 km between the measurement site and Dangxiong country, and separated by the Mt. Tanggula. In addition, previous studies have reported that high $NO_x$ production was due to photochemical reactions on the snow surface of the Tibetan Plateau (Fisher et al., 2005; Lin et al., 2021). Figure R11 also told us that the closest distance between the measurement site and the icy summit of Mt. Tanggula is only ~7.5 km. Therefore, we cannot exclude the possibility that $NO_2$ was transported from the Mt. Tanggula. $NO_2$ profile also showed a Gaussian shape, with maximum values appeared at 300-400 m layer. This is also a distinguishing feature of transport contributions for $NO_2$. Near-surface $NO_2$ concentrations at the measurement site were not high during the observation period.

[Figure]

Figure R11. Geographical location of CAS (NAMORS) and around environment.
Moreover, we also did WPSCF analysis (Figure R12), and the WPSCF passing through Mt. Tanggula showed high values at 300-400 m layer, especially at 400 m (> 0.3). It indicated that the important contribution to $NO_x$ from ice and snow on the top of Mt. Tanggula under strong ultraviolet radiation.

[Figure]

Figure R12. Spatial distributions of 24-h WPSCF values for $NO_2$ at (a) 300 m, and (b) 400 m height layers from 01 May to 09 July 2019 over CAS (NAMORS).

15. Line 240-243: The vertical distributions of $O_3$ in the lower tropospheric layer are complex over the Tibetan Plateau. Why does the manuscript describe the "exponential shape" derived from previous studies here? The "exponential shape" is not consistent with the relatively uniform vertical variation of $O_3$ in this study.

Re: Thanks for your great comments.

The purpose of describing the exponential shape of $O_3$ profiles reported in previous studies was to show that the shape of $O_3$ profiles in Nam Co observation was credible. The $O_3$ profiles showed in Figure R13 were all from Tibetan Plateau measured by satellite, $O_3$ lidar and ozonesonde, especially for Lahsa and Yangbajing, which were closed to Nam Co. The height ranges presented in each of the plots in Figure R13 are not consistent, but the $O_3$ profiles monitored by different means below 4.0 km basically show an approximate exponential shape. It is possible that the different heights in these plots of Figure R7 created a visual misjudgment for the reviewer. We apologize for this. Moreover, due to the limited vertical resolution of MAX-DOAS $O_3$ profiles, the limited degrees of freedom and the constraints of the radiative transfer model caused by $O_3$ profile retrieval algorithm, the detailed structure of $O_3$ at different vertical altitudes may be missing, but it essentially characterized the vertical distribution of the $O_3$ profile in Nam Co.

We have rewritten these sentences as "The vertical gradient of $O_3$ concentration was also not obvious, which was associated with its vertical mixing and photochemical production. As shown in Figure S5, the corresponding TROPOMI $O_3$ profiles in Nam Co and $O_3$ profiles measured by lidar and ozonesonde around Nam Co reported in several previous studies also exhibited an exponential shape".

[Figure]

Figure R13. Ozone vertical profile measure by (a) TROPOMI at Nam Co, (b) lidar at Yangbajing (Fang et al., 2019), (c) ozonesonde at Qaidam (Zhang et al., 2020), (d) lidar at Lhasa (Yu et al., 2022), and (e) MAX-DOAS in this study.

16. Line 251: Please add "$O_3$".
Re: Thanks for your great comments. We have modified it as "3.2 Vertical distributions of aerosol, $H_2O$, $NO_2$, HONO and $O_3$"

17. Line 254-255: Why are $NO_2$ vertical profiles "Gaussian" distribution rather than Lorentz or other peak distribution?
Re: Thanks for your great comments.
As shown in Figure R2 (a), the aerosol vertical profile retrieved from MAX-DOAS show the best agreement with lidar measurement result. However, lidar could more accurately characterize the thickness of the layer of aerosol with high values. It should be attributed to lidar having a higher vertical resolution than MAX-DOAS. The vertical resolution of MAX-DOAS and lidar are 100 m and 7.5 m, respectively. Due to the uncertainty of the algorithm, MAX-DOAS is more insensitive at high altitudes and the degrees of freedom is often only 3 to 5. It is similar to the retrieval of $NO_2$ vertical profile. ***Therefore, in addition to the true vertical distribution of $NO_2$, the factors limiting the shape of $NO_2$ vertical profile also include the limited vertical resolution***

***and the low degree of freedom of the vertical profiles***. The $NO_2$ profile showed a Gaussian shape during our observation.

Referred to our previous study (Xing et al., 2017), we defined a Lorentz a priori with equivalent aerosol loading compared to the corresponding lidar profile. We found that the aerosol profile retrieved from Lorentz a priori shows a significant overestimation compared to the lidar results between 1 and 2 km. Besides, the degree of freedom of signal (DFS) of using Gaussian a priori (2.96) is higher than that of using Lorentz a priori (2.4). The retrieved error of adopting Lorentz a priori is relatively higher than the Gaussian a priori at the lowest 1 km (see Figure R14). This also goes some way to proof of rationality of MAX-DOAS Gaussian profile compared to the Lorentz profile.

[Figure]

Figure R14. The retrieved error of using Gaussian and Lorentz a priori for aerosol retrieval.

18. Line 264-265: Please clarify the monsoon transport leading to the elevation of maximum $H_2O$ layer.

Re: Thanks for your great comments.

As shown in Figure R15, we could find the Indian monsoon gradually increased from June, the stronger monsoon lasted from early June until the end of September. But we could find that there was a significantly decrease for the Indian monsoon index in the early July. Our campaign was from 01 May to 09 July 2019. It could explain the elevation of the height layers of maximum water vapor in June.

[Figure]

Figure R15. The Indian monsoon index during 2019.

Moreover, we have rewritten this sentence as "*The monthly averaged vertical profile of*

*H₂O in May and July exhibited an exponential shape, while its maximum concentration layer slightly elevated to 0.1-0.2 km in June which was related to the strongest monsoon transport.*".

19. Line 279-282: With respect to the vertical profiles of atmospheric component at the height of 3-4 km, are the retrieved data valid? How to understand the differences of $O_3$ profiles between "exponential shape" (Line 282) and "relatively uniform vertical gradient" (Line 240)?

Re: Thanks for your great comments.

The averaging kernels at higher altitudes (> 2.0 km) were relatively lower than that at low altitudes. This could lead to a reduction in the sensitivity of pollutants' concentrations at high altitudes. The uncertainty of the concentrations of pollutants at high altitudes were also larger than that at lower altitudes. However, this did not mean that the profiles at 3-4 km were not credible. Several previous studies also carried out scientific research using MAX-DOAS profiles with a height range of 0.0-4.0 km (Wang et al., 2019; Hoque et al., 2022; Kuhn et al., 2024). The vertical profiles of aerosol and trace gases (0.0-4.0 km) retrieved from MAX-DOAS were validated with aircraft measurements and model simulations, and they all shown good agreement (Wang et al., 2019; Hoque et al., 2022; Kuhn et al., 2024). It indicated that the MAX-DOAS profiles at high altitude were credible. For $O_3$ vertical profile, since we used the TROPOMI $O_3$ profiles as a constraint, its sensitivity was stronger above 3km. In this study, we only validated the surface concentrations of pollutants during our measurements, considering the absence of other independent vertical observations. We focused on analyzing concentration variations of pollutants below the PBL.

There may be a misunderstanding like comment 12. Figure 4 depicted the time series of daily averaged concentrations of pollutants at five altitudes during the whole observations. Figure 5 showed the averaged diurnal variations pollutants. As shown in Figure 4, we could find that the ratios of $O_3$ at 0.4-0.6 km, 0.8-1.0 km, 1.2-1.4 km and 1.6-1.8 km to those at 0.0-0.2 km were 89.25%, 82.44%, 80.16% and 79.13%, respectively. Therefore, we concluded that the vertical gradient of daily averaged $O_3$ concentration was also not obvious. We have rewritten this sentence as "The vertical gradient of daily averaged $O_3$ concentration was also not obvious, which was associated with its vertical mixing and photochemical production".

20. Line 298-301: It just lists the possible influencing factors here. To what extent, can the weak surface wind affect the sand-raising process?

Re: Thanks for your great comments.

The diurnal variation showed a bi-peak pattern, which was in line with the investigation reported by Pokharel et al. (2019). The first peak should be attributed to the local emission of aerosol and the diurnal cycle of PBL (Zhang et al., 2017; Pokharel et al., 2019). The second peak was driven by regional transport and the interaction between local sandy silt loam surface and local meteorology. Figure R4 told us that the high wind speed (> 4.5 m/s) at surface appeared after 15:00, which coincided with the appearance of the second aerosol peak. As shown in Figure 5, the high extinction during

the second peak period was extended to 1.0 km. Figure R16 told us that the wind speed above 500 m was larger than 8 m/s, which created a favorable condition for high-altitude aerosol transport.

In addition, we also noticed that there would be drift blown up by the wind at the measurement site after 15:00 during the whole observation period.

[Figure]

Figure R16. Wind direction and wind speed at (a) 10 m, (b) 500 m, (c) 1000 m, (d) 1300 m, and (e) 1800 m at a range of 25ºN-35ºN and 85ºE-95ºE, respectively.

These sentences have been rewritten as "*The first peak should be attributed to the local emission of aerosol and the diurnal cycle of PBL (Zhang et al., 2017; Pokharel et al., 2019). The second peak was driven by regional transport and the interaction between local sandy silt loam surface and local meteorology. The high wind speed (> 4.5 m/s) at surface appeared after 15:00, which coincided with the appearance of the second aerosol peak (Figure S3). Moreover, the high extinction during the second peak was extended to 1.0 km associated with the wind speed larger than 8 m/s (Figure S7), which created a favorable condition for high-altitude aerosol transport.*".

21. Line 302-308: Please check the attribution of $H_2O$ and $NO_2$ variations. For example, the monsoon not only affect the $H_2O$ variation in the morning but also in the afternoon.
Re: Thanks for your great comments.
We couldn't agree with you more. There maybe some misunderstandings for the sentences of "*The first peak appeared between 08:00-12:00, which was mainly affected by the monsoon drived long-range transport of $H_2O$. The second and third peaks occurred at 15:00-16:00 and after 17:00, respectively. In addition to long-range transport, the enhanced evaporation from the Nam Co lake also significantly contributed to the appearance of these two peaks of $H_2O$.*". It means that these three peaks of $H_2O$ all affected by the monsoon drived long-range transport. In addition, the enhanced evaporation from the Nam Co lake also significantly contributed to the second and third $H_2O$ peaks.

For the source of NO₂, we have rewritten the corresponding sentences as "*NO₂ mainly distributed at 0.2-0.4 km, and peaked before 10:00 and after 18:00 which were dominated by the effects of local emissions and regional transport from the NOx formed through ice and snow on the top of Mt. Tanggula under strong ultraviolet radiation (Figure S5) (Boxe et al., 2005; Fisher 2005; Chen et al., 2019; Lin et al., 2021).*".
Detailed analysis could referred to comment 12.

22. Line 314-315: Please check that the text description is consistent with the figures.
Re: Thanks for your great comments.
We confirmed that the text description is consistent with Figure 5. In order to determine this fact, we analyzed the averaged aerosol, H₂O, NO₂ and HONO profiles during three HONO peak periods (Figure R17).

[Figure]

Figure R17. The averaged aerosol, H₂O, NO₂ and HONO profiles during three HONO

peak periods

23. Figure 7: The validity of results above 3 km should be carefully checked. Probably the concentrations of atmospheric component at higher height mainly reflected a priori information.

Re: Thanks for your great comments.

The averaging kernels at higher altitudes (> 2.0 km) were relatively lower than that at low altitudes. This could lead to a reduction in the sensitivity of pollutants' concentrations at high altitudes. The uncertainty of the concentrations of pollutants at high altitudes were also larger than that at lower altitudes. However, this did not mean that the profiles at 2-4 km were not credible. Several previous studies also carried out scientific research using MAX-DOAS profiles with a height range of 0.0-4.0 km (Wang et al., 2019; Hoque et al., 2022; Kuhn et al., 2024). The vertical profiles of aerosol and trace gases (0.0-4.0 km) retrieved from MAX-DOAS were validated with aircraft measurements and model simulations, and they all shown good agreement (Wang et al., 2019; Hoque et al., 2022; Kuhn et al., 2024). It indicated that the MAX-DOAS profiles at high altitude were credible. For $O_3$ vertical profile, since we used the TROPOMI $O_3$ profiles as a constraint, its sensitivity was stronger above 2 km. In this study, we only validated the surface concentrations of pollutants during our measurements, considering the absence of other independent vertical observations. We focused on analyzing concentration variations of pollutants below the PBL.

24. Line 388-390: Is that so?

Re: Thanks for your great comments.

This is our mistake. We added dashed line representing the scatter distribution envelope in Figure R18. We could find that the maximum value of $HONO/NO_2$ appeared around water vapor being around $1.0×10^{17}$ molec $cm^{-3}$ under 1.0 km, and being around $1.0-2.0×10^{17}$ molec $cm^{-3}$ at 0.5-1.0 km height layer.

[Figure]

Figure R18. Scatter plots between $HONO/NO_2$ and $H_2O$ colored by aerosol extinction at (a) 0.0-0.2 km, (b) 0.4-0.6 km, (c) 0.8-1.0 km, (d) 1.2-1.4 km, (and e) 1.6-1.8 km

from 1$^{st}$ May to 9$^{th}$ July 2019. The dashed line represents the scatter distribution envelope.

This sentence has been modified as "*We found that the maximum value of HONO/NO$_2$ appeared around water vapor being around 1.0×10$^{17}$ molec cm$^{-3}$ under 1.0 km, and being around 0.5-1.0×10$^{17}$ molec cm$^{-3}$ at 1.0-2.0 km height layer.*".

25. Figure 9: Please add the graphic symbol description, such as error bar. Please modify all the similar problems in other figures.

Re: Thanks for your great comments. We have made following modifications.

(1) Figure 9. Statistics for the vertical profile of HONO/NO$_2$ from 1$^{st}$ May to 9$^{th}$ July 2019. The left and right of the blue box represent the 25$^{th}$ and 75$^{th}$ percentiles, respectively; the dot within the box represent the mean.

(2) Figure 3. Averaged diurnal variation of AOD at CAS (NAMORS). The error bars represent the mean retrieved errors of AOD.

(3) Figure 6. Validations of (a) MAX-DOAS NO$_2$ vs in situ NO$_2$ (error bars represent the retrieved errors of NO$_2$ from MAX-DOAS and BBCES), (b) MAX-DOAS HONO vs LOPAP HONO, (c) MAX-DOAS O$_3$ vs in situ O$_3$, and (d) MAX-DOAS PBL vs WRF PBL.

26. Line 420: three clusters? There are four clusters in Figure S7e.

Re: Thanks for your great comments. We have made modifications as following:

"*As shown in Figure S9 and Table 4, the backward trajectories arriving at NAMORS during the observation were classified into three clusters at 200 m, 600 m, 1400 m, 1800 m, and four clusters at 1000 m. We found that cluster 3 was associated with the highest O$_3$ concentration at 200 m (65.48±17.41 ppb) and 1800 m (49.69±2.21 ppb), and cluster 1 were related to the highest O3 concentration at 600 m (54.67±6.94 ppb), 1000 m (51.61±3.84 ppb) and 1400 m (50.51±2.89 ppb).*"

*Table 4. Trajectory ratios and averaged O$_3$ concentration for all trajectory clusters arriving in Nam Co at 200 m, 600 m, 1000 m, 1400 m and 1800 m from May to July 2019.*

| | Cluster | Traj_ratio | O$_3$ concentration (ppb) Mean ±SD |
|---|---|---|---|
| 200 m | 1 | 55.86% | 61.50 ±18.15 |
| | 2 | 11.85% | 54.57 ±14.67 |
| | 3 | 32.28% | 65.48 ±17.41 |
| | All | 100.00% | 61.14 ±17.74 |
| 600 m | 1 | 62.55% | 54.67 ±6.94 |
| | 2 | 14.32% | 50.43 ±6.64 |
| | 3 | 23.13% | 53.27 ±7.63 |
| | All | 100.00% | 53.39 ±7.26 |
| 1000 m | 1 | 49.16% | 51.61 ±3.84 |
| | 2 | 8.81% | 49.60 ±3.99 |
| | 3 | 22.73% | 50.72 ±4.21 |
| | 4 | 19.30% | 51.39 ±4.49 |

| | | | |
|---|---|---|---|
| | *All* | *100.00%* | *50.98 ±4.30* |
| | *1* | *80.14%* | *50.51 ±2.89* |
| *1400 m* | *2* | *4.95%* | *49.12 ±2.73* |
| | *3* | *14.92%* | *49.44 ±3.85* |
| | *All* | *100.00%* | *50.07 ±3.15* |
| | *1* | *83.75%* | *49.68 ±2.55* |
| *1800 m* | *2* | *0.00%* | *49.07 ±2.23* |
| | *3* | *16.25%* | *49.69 ±2.21* |
| | *All* | *100.00%* | *49.59 ±2.49* |

27. Line 434-437: The discussion of stratospheric O$_3$ intrusion is too simple. The logic is confusing between stratosphere intrusion and long-range transport.

Re: Thanks for your great comments.

Several previous studies have revealed that the stratospheric O$_3$ intrusion events were frequent in the Himalayas during spring and summer. The O$_3$ from stratospheric intrusions in the Himalayas can affect the O$_3$ at NAMORS through long-range transport. Figure R19 showed that the contribution of O$_3$ transported from Himalayas can even up to 50 ppb, especially under 600 m.

[Figure]

Figure R19. Spatial distributions of WCWT values for O$_3$ at (a) 200 m, (b) 600 m, (c) 1000 m, (d) 1400 m, and (e) 1800 m height layers from 01st May to 09th July 2019 over CAS (NAMORS).

We can understand the process as following:

The WCWT showed that the Himalayas was a very important potential source area of O$_3$ at NAMORS. Therefore, we should also understand the source of O$_3$ in the Himalayas. Several previous studies have revealed that the stratospheric O$_3$ intrusions were an important source of O$_3$ in the Himalayas during spring and summer.

Theses sentences have been modified as "*In addition, Figure 10 showed that the*

*contribution of O₃ transported from Himalayas can even up to 50 ppb, especially under 600 m. Several previous studies have revealed that the stratospheric O₃ intrusion events were frequent in the Himalayas during spring and summer (Cristofanelli et al., 2010; Chen et al., 2011; Škerlak et al., 2014; Putero et al., 2016). Therefore, the O₃ from stratospheric intrusions in the Himalayas can affect the O₃ at NAMORS through long-range transport.*".

It should be noted that our present observational data do not capture the O₃ stratospheric intrusion process in the Himalaya. We also hope to follow this work in the future.

28. Section References (Line 482-854): Although there are too many references, the key references related to this topic are missing.

Re: Thanks for your great comments.

We have added some literature that supports our some conclusions in the study.

*Added references:*

[revised manuscript text omitted]

[22] Kuhn, L., Beirle, S., Kumar, V., Osipov, S., Pozzer, A., Bosch, T., Kumar, R., Wagner, T.: On the influence of vertical mixing, boundary layer schemes, and temporal emission profiles on tropospheric $NO_2$ in WRF-Chem-comparisons to in situ, satellite, and MAX-DOAS observations, Atmos. Chem. Phys., 2024, 24, 185-217, doi:10.5194/acp-24-185-2024.

[23] Hoque, H. M. S., Sudo, K., Irie, H., Damiani, A., Naja, M., Fatmi, A. M.: Multi-axis differential optical absorption spectroscopy (MAX-DOAS) observations of formaldehyde and nitrogen dioxide at three sites in Asia and comparison with the global chemistry transport model CHASER, Atmos. Chem. Phys., 2022, 22, 12559-12589,

doi:10.5194/acp-22-12559-2022.

[24] Wang, Y., Donner, S., Donner, S., Bohnke, S., De Smedt, I., Dickerson, R. R., Dong, Z., He, H., Li, Z., Li, Z., Li, D., Liu, D., Ren, X., Theys, N., Wang, Y., Wang, Y., Wang, Z., Xu, H., Xu, J., Wagner, T.: Vertical profiles of $NO_2$, $SO_2$, HONO, HCHO, CHOCHO and aerosols derived from MAX-DOAS measurements at a rural site in the central western North China Plain and their relation to emission sources and effects of regional transport, Atmos. Chem. Phys., 2019, 19, 5417-5449, doi:10.5194/acp-19-5417-2019.

[25] Pokharel, M., Guang, J., Liu, B., Kang, S., Ma, Y., Holben, B.N., Xia, X., Xin, J., Ram, K., Rupakheti, D., Wan, X., Wu, G., Bhattarai, H., Zhao, C., and Cong, Z.: Aerosol properties over Tibetan Plateau from a decade of AERONET measurements: Baseline, types, and influencing factors, J. Geophys. Res.: Atmos., 124, 13357-13374, doi:10.1029/2019JD031293, 2019.

[26] Wang, J., Zhang, Y., Zhang, C., Wang, Y., Zhou, J., Whalley, L. K., Slater, E. J., Dyson, J. E., Xu, W., Cheng, P., Han, B., Wang, L., Yu, X., Wang, Y., Woodward-Massey, R., Lin, W., Zhao, W., Zeng, L., Ma, Z., Heard, D. E., Ye, C.: Validating HONO as an intermediate tracer of the external cycling of reaction nitrogen in the background atmosphere. Environ. Sci. Technol., 57, 13, 5474-5484, doi:10.1021/acs.est.2c06731, 2023.

[27] Xia, X., Zong, X., Cong, Z., Chen, H., Kang, S., and Wang, P.: Baseline continental aerosol over the central Tibetan plateau and a case study of aerosol transport from South Asia, Atmos. Environ., 45, 7370-7378, doi: 10.1016/j.atmosenv.2011.07.067, 2011.

[28] He, S., Wang, S., Zhang, S., Zhu, J., Sun, Z., Xue, R., Zhou, B.: Vertical distributions of atmospheric HONO and the corresponding OH radical production by photolysis at the suburb area of Shanghai, China. Sci. Total Environ., 858, 159703, doi:10.1016/j.scitotenv.2022.159703, 2023.

---

## Referee Report (RR1)

The authors have provided a detailed response to the original two referees and in particular the error budget, underlying DOAS fits, and TUV model set up to determine OH production are now much more clear. They have also provided further support for near surface ozone and transport of ozone and $NO_x$ to the site. These provide good support for some of their key results near the surface which are most results reported in the abstract. However, questions particular to the vertical profiles remain which could indirectly still impact the key results.

I have two major comments:

1)  What is the information content of the retrieved profiles? Can the authors provide representative or average values for averaging kernels (AVK) or degrees of freedom (DoF) for specific altitudes? The authors report that results with less than 1 DoF are filtered, but this will typically be concentrated near the surface. In Fig. 4 results are shown as high as 1.8 km agl and in in Figs. 5 and 7 to 4 km agl. Are the results at higher altitudes significant or simply conforming to the a priori? Information is needed to assess this.

2)  More information is still needed regarding the calculation of and reporting of results related to OH production.

    a.  Firstly, the authors state in the abstract "$O_3$ and HONO were the main contributors to OH on the TP" and have similar language to this effect in the main text, however, they do not appear to consider other sources. This is therefore not a finding and the language should reflect that.

    b.  Related to major comment 1, $O_3$ and $H_2O$ appear to be sometimes or always elevated at high altitudes relative to other retrieved species potentially driving the resulting OH source.

        i.   $O_3$ appears to never drop below 48 ppb, is this based on the a priori or is it retrieved? If it is from the a priori why is $O_3$ given a non-zero concentration it decays to when other species are not. Is it based on TROPOMI or lidar data in Fig. S4 if so that needs to be explicit? These would seem to leave room for substantial variability in the free tropospheric background. I will also note that the various traces in Fig. S4 are not explained and units are not the same across panels.

        ii.  $H_2O$ appears to frequently increase above ~3 km agl, sometimes to concentrations greater than at the surface despite presumably lower temperatures and pressures at those altitudes. Is this allocation of the remaining column being placed at high altitudes or actually localized at the higher altitudes?

        iii. The TP is a frequent site of stratospheric intrusions (Škerlak et al., 2015) as has been observed at Nam Co in particular (Yin et al., 2017) one would expect this to drive greater profile variability for $O_3$

and $H_2O$ (and possibly other gases) is this hiding in the averages or is there a reason it is not detected? If there are high-$O_3$ and low-$H_2O$ air masses descending over the site will that impact the retrieval?

c. The authors have provided a detailed response regarding their implementation of TUV already, but can they address how uncertainty in the retrieved profiles might impact the TUV calculations?

References:

Škerlak, B., Sprenger, M., Pfahl, S., Tyrlis, E., and Wernli, H.: Tropopause folds in ERA-Interim: Global climatology and relation to extreme weather events, Journal of Geophysical Research: Atmospheres, 120, 4860–4877, https://doi.org/10.1002/2014JD022787, 2015.

Yin, X., Kang, S., De Foy, B., Cong, Z., Luo, J., Zhang, L., Ma, Y., Zhang, G., Rupakheti, D., and Zhang, Q.: Surface ozone at Nam Co in the inland Tibetan Plateau: Variation, synthesis comparison and regional representativeness, Atmos Chem Phys, 17, 11293–11311, https://doi.org/10.5194/ACP-17-11293-2017, 2017.

---

## Author Response (AR2)

**Point-to-point responses**

*We appreciate the reviewers for their valuable and constructive comments, which are very helpful for the improvement of the manuscript. We have revised the manuscript carefully according to the reviewers' comments. We have addressed the reviewers' comments on a point-to-point basis as below for consideration, where the reviewers' comments are cited in **black**, and the responses are in **blue**.*

**Referee #1**

The authors have provided a detailed response to the original two referees and in particular the error budget, underlying DOAS fits, and TUV model set up to determine OH production are now much more clear. They have also provided further support for near surface ozone and transport of ozone and $NO_x$ to the site. These provide good support for some of their key results near the surface which are most results reported in the abstract. However, questions particular to the vertical profiles remain which could indirectly still impact the key results.

I have two major comments:
1. What is the information content of the retrieved profiles? Can the authors provide representative or average values for averaging kernels (AVK) or degrees of freedom (DOF) for specific altitudes? The authors report that results with less than 1 DOF are filtered, but this will typically be concentrated near the surface. In Fig. 4 results are shown as high as 1.8 km agl. and in in Figs. 5 and 7 to 4 km agl. Are the results at higher altitudes significant or simply conforming to the a priori? Information is needed to assess this.

Re. Many thanks for your great comments.

(1) The information content for vertical profile quality control includes: cloud information, retrieval error, averaging kernel (AVK, sensitivity to different altitudes), degrees of freedom (DOF, trace of the averaging kernel matrix), and cost function. The information content for the retrieved profiles include: vertical profiles, averaging kernels, gain, retrieval errors, weighting function, cost function (Chisquare), degree of freedom for signal, retrieved VCDs and the corresponding errors.

(2) The average averaging kernels (AVK) and degree of freedom (DOF) for aerosol, $H_2O$, $NO_2$, HONO and $O_3$ were shown in Figure R1.

[Figure]

Figure R1. The average AVK and DOF of the retrieved aerosol (a), $H_2O$ (b), $NO_2$ (c), HONO (d) and $O_3$ (e), respectively.

The average AVK and DOF in our study is at the same level as previous studies (Bosch et al., 2018; Friess et al., 2019). AVK denotes the sensitivity of the retrieval at different heights and DOF denotes the trace of the AVK matrix. From the above results, we can also find that the retrieval results have sensitivity at high altitude, even 4 km.

(3) In our study, we used a combination of observation and gain iterations, net of the algorithm's dependence on a priori profile. Therefore, the retrieved results are independent of the a priori profiles, and it only act as an intermediate variable in the iterative process.

(4) In this study, we estimated the contribution of different error sources to the AOD and VCDs of trace gases, and near-surface (0–200 m) trace gases' concentrations and aerosol extinction coefficients (AECs), respectively. The detailed demonstrations and estimation methods are displayed below, and the corresponding varies errors are summarized in Table 1.

a.  Smoothing errors arise from the limited vertical resolution of profile retrieval. Noise errors denote the noise in the spectra (i.e., the error of DOAS fits). Considering the error of the retrieved state vector equaling the sum of these two independent errors, we calculated the sum of smoothing and noise errors on near-surface concentrations and column densities, which were 13 and 5 % for aerosols, 13 and 36 % for $H_2O$, 12 and 14 % for $NO_2$, 18 and 21 % for HONO, and 12 and 32 % for $O_3$, respectively.

b.  Algorithm error is denoted by the differences between the measured and simulated DSCDs. This error contains forward model error from an imperfect approximation of forward function, parameter error of forward model, and other errors, such as detector noise (Rodgers, 2004). Algorithm error is a function of the viewing angle, and it is difficult to assign this error to each altitude. Thus, this error on the near-surface values and column densities is estimated through calculating the average relative differences between the measured and simulated DSCDs at the minimum and maximum elevation angle (except 90°), respectively (Wagner et al., 2004). In this study, we estimated these errors on the near-surface values and the column densities at 4 and 8 % for aerosols, 3 and 11 % for $NO_2$, and 20 and 20 % for HONO referring to Wang et al. (2017, 2020), 1 and 8 % for $H_2O$ referring to Lin et al. (2020), and 6 and 10 % for $O_3$ referring to Ji et al. (2023), respectively.

c.  Cross section error arises from the uncertainty in the cross section. According to Thalman and Volkamer, (2013), Lin et al. (2020), Vandaele et al. (1998), Stutz et al. (2000), and Serdyuchenko et al. (2014), we adopted 4, 3, 3, 5, and 2 % for $O_4$ (aerosols), $H_2O$, $NO_2$, HONO and $O_3$, respectively.

d.  The profile retrieval error for trace gases is sourced from the uncertainty of aerosol extinction profile retrieval and propagated to trace gas profile. This error could be roughly estimated based on a linear propagation of the total error budgets of the aerosol retrievals. The errors of the learned four trace gases were roughly estimated at 14 % for VCDs and 10 % for near-surface concentrations, respectively.

The total uncertainty was the sum of all above errors in the Gaussian error propagation, and the error results were listed in Table 1. We found that the smoothing and noise errors played a dominant role in the total uncertainties of aerosol and trace gases. Moreover, improving the accuracy and temperature gradient of the absorption cross

section is another important means to reduce the uncertainty of the vertical profiles in the future, especially for $O_3$.

Table 1. Error budget estimation (in %) of the retrieved near-surface (0–200 m) concentrations of trace gases and AECs, and AOD and VCDs.

| | | Error sources | | | | Total |
|---|---|---|---|---|---|---|
| | | Smoothing and noise errors | Algorithm error | Cross section error | Related to the aerosol retrieval (only for trace gases) | |
| Near-surface | aerosol | 13 | 4 | 4 | - | 14 |
| | $H_2O$ | 13 | 1 | 3 | 14 | 19 |
| | $NO_2$ | 12 | 3 | 3 | 14 | 18 |
| | HONO | 18 | 20 | 5 | 14 | 29 |
| | $O_3$ | 12 | 6 | 2 | 14 | 19 |
| VCD or AOD | AOD | 5 | 8 | 4 | - | 10 |
| | $H_2O$ | 36 | 8 | 3 | 10 | 38 |
| | $NO_2$ | 14 | 11 | 3 | 10 | 20 |
| | HONO | 21 | 20 | 5 | 10 | 31 |
| | $O_3$ | 32 | 10 | 2 | 10 | 35 |

Certainly, we also did independent validations. However, there was not other vertical observations during our campaign on the TP. Therefore, we did validations between in situ measurements and the bottom layer of MAX-DOAS profiles (Figure R2).

[Figure]

Figure R2. Validations of (a) MAX-DOAS $NO_2$ vs in situ $NO_2$, (b) MAX-DOAS HONO vs LOPAP HONO, (c) MAX-DOAS $O_3$ vs in situ $O_3$.

Moreover, we did vertical-profile validations in Shanghai and Beijing. As shown in Figure R3, we used Mie lidar to validate MAX-DOAS aerosol vertical profiles, and used balloon-based $NO_2$ profiles to validate MAX-DOAS $NO_2$ vertical profiles (Xing et al., 2017). The good agreement indicates the reliability of MAX-DOAS retrieved aerosol and $NO_2$ profiles.

[Figure]

Figure R3. Validations of (a) MAX-DOAS aerosol profile vs lidar aerosol profile, (b) MAX-DOAS $NO_2$ profile vs balloon-based $NO_2$ profile.

(5) Finally, the intercomparison of HONO results during CINDI-2 campaign reported by Thomas Wagner's group revealed "Another interesting finding for the "EnviMes" instruments is that although the same set of spectra measured by the "USTC" instruments is analysed by the "DLR" and "USTC" researchers, much larger rms values and fit errors are found for the "DLR(1)" and "DLR(2)" results (especially for the "DLR(2) " results with the "sequential FRS") than for the "USTC(1)" and "USTC(2)". (Wang et al., 2020)"

2. More information is still needed regarding the calculation of and reporting of results related to OH production.
a. Firstly, the authors state in the abstract "$O_3$ and HONO were the main contributors to OH on the TP" and have similar language to this effect in the main text, however, they do not appear to consider other sources. This is therefore not a finding and the language should reflect that.
Re. Many thanks for your great comments.
As reported in previous studies, the precursors of OH on the Tibetan Plateau were $O_3$, HONO, $NO_2$ and HCHO (Lin, et al., 2008; Xing, et al., 2021; Lyu, et al., 2020; Zhang, et al., 2021; Wang, et al., 2023). While considering the low background concentration of $NO_2$ and HCHO, and their pathways to produce OH, the contribution to OH of $NO_2$ and HCHO than HONO and $O_3$ is lower (Sörgel, et al., 2011).
Moreover, we have modified our description as following:
(1) That indicated $O_3$ was an important contributor of OH production (> 80%) on the TP, which was about 5-6 times to HONO.
(2) $O_3$ and HONO were important source of OH on the TP.

b. Related to major comment 1, $O_3$ and $H_2O$ appear to be sometimes or always elevated at high altitudes relative to other retrieved species potentially driving the resulting OH source.
i. $O_3$ appears to never drop below 48 ppb, is this based on the a priori or is it retrieved? If it is from the a priori why is $O_3$ given a non-zero concentration it decays to when other species are not. Is it based on TROPOMI or lidar data in Fig. S4 if so that needs to be explicit? These would seem to leave room for substantial variability in the free tropospheric background. I will also note that the various traces in Fig. S4 are not explained and units are not the same across panels.
Re. Many thanks for your great comments.
The vertical profiles in Figure 5 (e) of the manuscript were averaged profiles. Figure R4 showed the hourly variations of surface ozone during all the observation period. The minimum and maximum concentrations of the surface $O_3$ were 29.65 ppb and 162.37 ppb, respectively.

[Figure]

Figure R4. Variations of surface $O_3$ concentrations from 27 April to 09 July 2019.

Another fact that we need to account for is that we only observed $O_3$ variations during the daytime. For the retrieval algorithm, aiming at the bottleneck of strong absorption interference in the stratosphere, which makes it difficult to realize $O_3$ profile retrieval by ground-based MAX-DOAS alone, a joint satellite-ground based hyperspectral remote sensing algorithm was developed. Reduction of stratospheric $O_3$ absorption interference by 80-90% by coupling hyperspectral satellite remote sensing of stratospheric $O_3$ profiles. Moreover, we have solved the problem of insufficient sensitivity of ground-based MAX-DOAS to high-altitude $O_3$ by coupling satellite remote sensing tropospheric $O_3$ observations to build an inverse a priori profile dataset (Ji et al., 2023). Therefore, the retrieved results are independent of the a priori profiles, and it only act as an intermediate variable in the iterative process. We have also validated the retrieved O3 profiles with tower-based in-situ measurements (Figure R5).

[Figure]

Figure R5. Linear regression plots for $O_3$ comparison results at different altitudes; IAP site (a) MAX-DOAS 0-100 m layer vs 2 m in tower; (b) MAX-DOAS 0-100 m layer vs 60 m in tower; (c) MAX-DOAS 100-200 m layer vs 160 m in tower; (d) MAX-

DOAS 200-300 m layer vs 280 m in tower.

The retrieved results depended on the true concentration of atmospheric species in the atmosphere and the corresponding detection limits of the instrument. The detection limit can be quantified as following:

$$\overline{D}_{\mathrm{lim}\mathit{it}} \approx \sigma \times \frac{6}{\sqrt{n-1}}$$

Where, $n$ and $\sigma$ were the number of channels and noise level of the spectrometer, respectively. The detection limits of aerosol, $H_2O$, $NO_2$, HONO and $O_3$ were shown in Table 2.

Table 2. Detection limit of our instrument.

|  | Detection limits |
|---|---|
| Aerosol | 0.05 km$^{-1}$ |
| H$_2$O | 0.05% |
| NO$_2$ | 0.1 ppb |
| HONO | 0.1 ppb |
| O$_3$ | 0.1 ppb |

Figure S4 depicted the diurnal variation of PBL height, which was calculated using aerosol vertical profiles to calculate the heights with fastest variation rates of aerosols. We have also validated the calculated PBL with WRF simulated PBL.

Figure S8 showed the $O_3$ vertical profiles measured by TROPOMI at Nam Co, lidar at Yangbajing, ozonesonde at Qaidam, and lidar at Lhasa. These data and figures were all from previous researches. The purpose of referring these studies was only to elucidate that the exponential decreasing vertical profile shape of $O_3$ below 1 km in the Nam Co region is reasonable.

ii. $H_2O$ appears to frequently increase above ~3 km agl, sometimes to concentrations greater than at the surface despite presumably lower temperatures and pressures at those altitudes. Is this allocation of the remaining column being placed at high altitudes or actually localized at the higher altitudes?

Re. Many thanks for your great comments.

Firstly, we only retrieved the vertical profiles on sunny and cloud-free periods. The influence of cloud can be filtered.

Secondly, the accuracy of the $H_2O$ vertical profile algorithm has been validated at another station (Beijing) as following.

Three representative water vapor profiles were retrieved from the MAX-DOAS measurements taken on three clear days: (a) 0 May 2018; (b) 19 June 2018; and (c) 31 July 2018; these were validated with the corresponding ballon-borne radiosonde profiles, as shown in Figure R6. The surface concentrations of these representative profiles are located in different concentration ranges. The ballon-borne radiosonde data were interpolated onto the MAX-DOAS grid for comparison. Furthermore, the interpolated ECMWF ERA-interim profiles were also used to validate the MAX-DOAS profiles. The correlation analysis results for the MAX-DOAS profiles and corresponding balloon-borne radiosonde profiles are displayed in the bottom panels of Figure R6, where the colors represent the height of each layer and horizontal gray lines

indicate the errors of retrieved profile in different height layers. The biases and standard deviations between MAX-DOAS profiles and corresponding balloon-borne radiosonde profiles on 8 May 2018; 19 June 2018; and 31 July 2018 are $-0.14\pm1.78\times10^{16}$, $-0.51\pm3.10\times10^{16}$, and $-1.10\pm4.36\times10^{16}$. Here, all values are in units of molec/cm$^3$. Overall, the water vapor profiles retrieved from MAX-DOAS and radiosonde measurements exhibit a high level of consistency, with high Pearson correlation coefficients (R) for these three profiles.

[Figure]

Figure R6. Three typical water vapor profiles retrieved from MAX-DOAS with the corresponding balloon-borne radiosonde measurements, ECMWF ERA-interim datasets, and NCDC in-situ measurements taken on 8 May 2018 (a); 19 June 2018 (b); and 31 July 2018 (c). The blue lines represent the water vapor concentration profiles measured by balloon-borne radiosonde. The pink lines and the shaded areas represent the MAX-DOAS retrieved water vapor profiles and their errors. The orange lines represent the water vapor profiles derived from the ECMWF ERA-interim reanalysis dataset. The dashed gray lines represent the a priori profile used in water vapor profile retrieval. The green dots represent the surface concentration of water vapor measured by the NCDC in-situ instruments. The bottom panels display the corresponding correlation analysis results between the MAX-DOAS-retried and radiosonde water vapor profiles on 8 May (a); 19 June 2018 (b); and 31 July 2018 (f), where colors indicate the height of each layer and the horizontal gray lines indicate the errors of retrieved profile in different height layers.

As shown in Figure R7, we conducted a correlation analysis of the water vapor concentrations in different height layers (derived from the MAX-DOAS and ECMWF ERA-interim data) to validate the profiles. As performed for the balloon-borne radiosonde measurements, the ECMWF ERA-interim water vapor profiles were also interpolated onto the MAX-DOAS grid. During this observation period, the MAX-DOAS instruments collected spectra from 00:00 to 10:00 (UTC). However, the

temporal resolution of the ECMWF ERA-interim dataset is 6 h. Thus, only the MAX-DOAS profiles measured at 00:00 and 06:00 can be used to validate the corresponding ECMWF profiles. These validations were conducted under no-cloud conditions by synchronizing the timetable to the AERONET data. In total, 138 profiles measured with MAX-DOAS in this observation period from 18 April to 30 September 2018 were validated using the coincident ECMWF profiles.

[Figure]

Figure R7. Correlation analysis of MAX-DOAS water vapor concentrations and ECMWF results in different the MAX-DOAS grid using a linear method to facilitate vertical layers. The 00:00 and 06:00 UTC ECMWF profiles are interpolated onto the comparison. Both the MAX-DOAS and ECWMF profiles are normalized to the AERONET timetable for cloud screening. The Pearson correlation coefficient (R), linear fitting slope (Slope), the bias with stand deviation (SD) between MAX-DOAS and ECMWF result as each layer are given in this figure. All values here are in units of $10^{17}$ molec/cm$^2$.

Good agreement between the MAX-DOAS and ECMWF results can be observed in height layers below 2000 m, with the Pearson correlation coefficient (R) ranging from 0.695 to 0.857. Under an increase in layer heigh, the detection sensitivity of MAX-DOAS gradually decreases and the correlation analysis results degrade. In the layer height from 600 to 1200 m, the consistency between the MAX-DOAS concentrations and the ECMWF results in slightly worse, which may be the result of the large uncertainties in these layers. The decreasing detection sensitivity and the enhances constraint of a priori profile together lead to the decreasing linear fitting slope with

increasing height. MAX-DOAS results have high sensitivities in the lower atmosphere and the contribution of a priori profile in retrieved profile is relatively small. With the increasing height, the detection sensitivity gets worse and the dependence of retrieved profile in a priori profile becomes strong gradually. The fixed exponentially decreasing a priori profile with a surface concentration of $4.6×10^{17}$ molec/cm$^3$ and a scale height of 1.9 km may be too high to represent the water vapor concentration in high altitudes. Together with the low sensitivity in high altitudes, the retrieved concentrations in high altitudes can be higher than actual situation, thus the linear fitting slope decreases and the bias between MAX-DOAS and ECMWF increases, as shown in Figure R7.

Zhang et al. (2013) also reported that the water vapor concentration gradually increases from low to high altitudes below 6000 m, and peaks at 5000 m-6000 m, during the Indian Ocean monsoon (Figure R8). Moreover, the water vapor transport direction is from southwest to northeast, which corresponds to the direction of the Indian Ocean monsoon.

[Figure]

Figure R8. Water vapor flux (streamlines; kg m$^{-1}$ s$^{-1}$) and divergence of moisture flux (contours; $10^{-5}$ kg m$^{-2}$ s$^{-1}$) for (top) 500-700 hPa and (middle) 300-500 hPa during the warm season.

Therefore, we believe that high-altitude water vapor on the Tibetan Plateau during the observation period is plausible.

iii. The TP is a frequent site of stratospheric intrusions (Škerlak et al., 2015) as has been observed at Nam Co in particular (Yin et al., 2017) one would expect this to drive greater profile variability for O$_3$ and H$_2$O (and possibly other gases) is this hiding in the averages or is there a reason it is not detected? If there are high-O$_3$ and low H$_2$O air masses descending over the site will that impact the retrieval?

Re. Many thanks for your great comments.

The spatial resolution of vertical profiles was 100 m. The temporal resolution of vertical profiles was less than 15 min. Based on this spatiotemporal resolution, it is possible to capture pollutants if they are transported from high altitude to near surface. Moreover, different pollutants have different spectral absorption structures (absorption cross sections), and in our algorithm there is little interaction between each species.

In this study, we can't find that $O_3$ transport from high altitude to ground surface. But, we found the transport process of $H_2O$ at higher altitudes, such as 17 May, 18 May, 23 May, and 30 June 2019 (Figure R9).

[Figure]

Figure R9. Diurnal variations of vertical profiles of $H_2O$ on 17 May, 18 May, 23 May, and 30 June 2019.

c. The authors have provided a detailed response regarding their implementation of TUV already, but can they address how uncertainty in the retrieved profiles might impact the TUV calculations?

Re. Many thanks for your great comments.

Previous studies reported the that the simulated actinic flux was larger 10%-40% than the measured actinic flux on clear and cloud-free days (Koepke et al., 1998; Badosa et al., 2005; Palancar et al., 2013; Ryu et al., 2017).

In this study, we used online TUV model.

*https://www2.acom.ucar.edu/modeling/tropospheric-ultraviolet-and-visible-tuv-radiation-model*

The main parameters for this model were cloud information, total ozone column, aerosol optical depth (AOD), single scattering albedo (SSA), and Ångström exponents. In this study, we selected clean and cloud free days, the AOD at 361 nm was derived from aerosol extinction profiles measured by MAX-DOAS; the daily total ozone column density was measured by TROPOMI with a value range of 260-280 DU; the single scattering albedo (SSA) was calculated based on the regression analysis of multi-wavelength (361 and 477 nm) $O_4$ absorptions measured by MAX-DOAS (Xing et al., 2019); fixed Ångström exponents of 0.508, 0.581 and 0.713 were used in May, June and July, respectively, referring to Xia et al. (2011).

The parameter scheme and the corresponding code of TUV model can be found at:
*https://www2.acom.ucar.edu/modeling/tuv-download*
As shown in Figure R10, when we put cloud information, total ozone column, aerosol optical depth (AOD), single scattering albedo (SSA), and Ångström exponents into the TUV model, the uncertainty of simulated actinic flux can decreased significantly (5%-9%).

[Figure]

Figure R10. |Measured actinic flux-TUV simulated actinic flux|×100%
In addition, the calculated OH were validated with measured OH, but the specific discussion will be organized in a separate study.

**References:**

[1] Škerlak, B., Sprenger, M., Pfahl, S., Tyrlis, E., and Wernli, H.: Tropopause folds in ERAInterim: Global climatology and relation to extreme weather events, Journal of Geophysical Research: Atmospheres, 120, 4860–4877, https://doi.org/10.1002/2014JD022787, 2015.

[2] Yin, X., Kang, S., De Foy, B., Cong, Z., Luo, J., Zhang, L., Ma, Y., Zhang, G., Rupakheti, D., and Zhang, Q.: Surface ozone at Nam Co in the inland Tibetan Plateau: Variation, synthesis comparison and regional representativeness, Atmos Chem Phys, 17, 11293– 11311, https://doi.org/10.5194/ACP-17-11293-2017, 2017.

[3] Lin, H., Liu, C., Xing, C., Hu, Q., Hong, Q., Liu, H., Li, Q., Tan, W., Ji, X., Wang, Z., and Liu, J.: Validation of water vapor vertical distributions retrieved from MAX-DOAS over Beijing, China, Remote Sens., 12, 3193, https://doi.org/10.3390/rs12193193, 2020.

[4] Zhang, Y., Wang, D., Zhai, P., Gu, G., and He, J.: Spatial distributions and seasonal variations of tropospheric water vapor content over the Tibetan Plateau, J. Climate, 26, 15, 5637-5654, https://doi.org/10.1175/JCLI-D-12-00574.1, 2013.

[5] Ji, X., Liu, C., Wang, Y., Hu, Q., Lin, H., Zhao, F., Xing, C., Tang, G., Zhang, Q., and Wagner, T.: Ozone profiles without blind area retrieved from MAX-DOAS measurements and comprehensive validation with multi-platform observations, Remote Sens. Environ., 284, 113339, https://doi.org/10.1016/j.rse.2022.113339, 2023.

[6] Lin, W., Zhu, T., Song, Y., Zou, H., Tang, M., Tang, X., and Hu, J.: Photolysis of surface $O_3$ and production potential of OH radicals in the atmosphere over the Tibetan Plateau, J. Geophys. Res.-Atmos., 113, D02309, doi:10.1029/2007JD008831, 2008.

[7] Xing, C., Liu, C., Wu, H., Lin, J., Wang, F., Wang, S., and Gao, M.: Ground-based vertical profile observations of atmospheric composition on the Tibetan Plateau (2017-2019), Earth Syst. Sci. Data, 13, 4897-4912, https://doi.org/10.5194/essd-13-4897-2021, 2021.

[8] Lyu, X., Guo, H., Zhang, W., Cheng, H., Yao, D., Lu, H., Zhang, L., Zeren, Y., Liu, X., Qian, Z., and Wang, S.: Anthropogenic emissions dominate ozone production in a high-elevation and highly forested region in central China: implications on forest ecosystems and regional air quality, ESS Open Active, doi:10.1002/essoar.10504327.1, 2020.

[9] Zhang, Y., Ju, T., Shi, Y., Wang, Q., Li, F., and Zhang, G.: Analysis of spatiotemporal variation of formaldehyde column concentration in Qinghai-Tibet Plateau and its influencing factors, Environ. Sci. Pollut. Res., 28, 55233-55251, doi:10.1007/s11356-021-14719-3, 2021.

[10] Wang, J., Zhang, Y., Zhang, C., Wang, Y., Zhou, J., Whalley, L. K., Slater, E. J., Dyson, J. E., Xu, W., Cheng, P., Han, B., Wang, L., Yu, X., Wang, Y., Woodward-Massey, R., Lin, W., Zhao, W., Zeng, L., Ma, Z., Heard, D. E., and Ye, C.: Validating HONO as an intermediate tracer of the external cycling of reactive nitrogen in the background atmosphere, Environ. Sci. Technol., 57, 5474-5484, doi:10.1021/acs.est.2c06731, 2023.

[11] Sörgel, M., Regelin, E., Bozem, H., Diesch, J. M., Drewnick, F., Fischer, H., Harder,

H., Held, A., Hosaynali-Beygi, Z., Martinez, M., and Zetzsch, C.: Quantification of the unknown HONO daytime source and its relation to $NO_2$, Atmos. Chem. Phys., 11, 20, 10433-10447, doi:105194/acp-11-10433-2011, 2011.

[12] Wang, Y., Apituley, A., Bais, A., Beirle, S., Benavent, N., Borovski, A., Bruchkouski, I., Chan, K. L., Donner, S., Drosoglou, T., Finkenzeller, H., Friedrich, M. M., Friess, U., Garcia-Nieto, D., Gomez-Martin, L., Hendrick, F., Hilboll A., Jin, J., Johnston, P., Koenig, T. K., Kreher, K., Kumar, V., Kyuberis, A., Lampel. J., Liu, C., Liu, H., Ma, J., Polyansky, O. L., Postylyakov, O., Querel, R., Saiz-Lopez, A., Schmitt, S., Tian, X., Tirpitz, J-L., Van Roozendael, M., Volkamer, R., Wang, Z., Xie, P., Xing, C., Xu, J., Yela, M., Zhang, C., and Wagner, T.: Inter-comparison of MAX-DOAS measurements of tropospheric HONO slant column densities and vertical profiles during the CINDI-2 campaign, Atmos. Meas. Tech., 13, 5087-5116, doi:10.5194/amt-13-5087-2020, 2020.

[13] Friess, U., Beirle, S., Bonilla, L. A., Bosch, T., Friedrich, M. M., Hendrick, F., Piters, A., Richter, A., van Roozendael, M., Rozanov, V. V., Spinei, E., Tirpitz, J-L., Vlemmix, T., Wagner, T., and Wang, Y.: Intercomparison of MAX-DOAS vertical profile retrieval algorithms: studies using synthetic data, Atmos. Meas. Tech., 12, 2155-2181, doi:10.5194/amt-12-2155-2019, 2019.

[14] Bosch, T., Rozanov, V., Richter, A., Peters, E., Rozanov, A., Wittrock, F., Merlaud, A., Lampel, J., Schmitt, S., de Hajj, M., Berkhout, S., Henzing, B., Apituley, A., den Hoed, M., Vonk, J., Tiefengraber, M., Muller, M., and Burrows, J. P.: BOREAS-a new MAX-DOAS profile retrieval algorithm for aerosols and trace gases, Atmos. Meas. Tech., 11, 6833-6859, doi:105194/amt-11-6833-2018, 2018.

[15] Aliwell, S. R., Van Roozendael, M., Johnston, P. V., Richter, A., Wagner, T., Arlander, D. W., Burrows, J. P., Fish, D. J., Jones, R. L., Tørnkvist, K. K., Lambert, J. C., Pfeilsticker, K., and Pundt, I.: Analysis for BrO in zenith-sky spectra: an intercomparison exercise for analysis improvement, J. Geophys. Res., 107, ACH 10-1–ACH 10-20, https://doi.org/10.1029/2001JD000329, 2002.

[16] Vandaele, A. C., Hermans, C., Simon, P. C., Carleer, M., Colin, R., Fally, S., Mérienne, M. F., Jenouvrier, A., and Coquart, B.: Measurements of the $NO_2$ absorption cross section from 42000 cm$^{-1}$ to 10000 cm$^{-1}$ (238–1000nm) at 220 K and 294 K, J. Quant. Spectrosc. Ra., 59, 171–184, 1998.

[17] Meller, R. and Moortgat, G. K.: Temperature dependence of the absorption cross sections of formaldehyde between 223 and 323 K in the wavelength range 225–375 nm, J. Geophys. Res., 105, 7089–7101, 2000.

[18] Volkamer, R., Spietz, P., Burrows, J., Platt, U.: High-resolution absorption cross-section of glyoxal in the UV-vis and IR spectral ranges, J. Photochem. Photobiol. A Chem., 172, 35–46, 2005.

[19] Rothman, L. S., Gordon, I. E., Barbe, A., Benner, D. C., Bernath, P. E., Birk, M., Boudon, V., Brown, L. R., Campargue, A., Champion, J. P., Chance, K., Coudert, L. H., Dana, V., Devi, V. M., Fally, S., Flaud, J. M., Gamache, R. R., Goldman, A., Jacquemart, D., Kleiner, I., Lacome, N., Lafferty, W. J., Mandin, J. Y., Massie, S. T., Mikhailenko, S. N., Miller, C. E., Moazzen-Ahmadi, N., Naumenko, O. V., Nikitin, A. V., Orphal, J., Perevalov, V. I., Perrin, A., Predoi-Cross, A., Rinsland, C. P., Rotger, M., Simeckova,

M., Smith, M. A. H., Sung, K., Tashkun, S. A., Tennyson, J., Toth, R. A., Vandaele, A. C., Vander Auwera, J.: The HITRAN 2008 molecular spectroscopic database, J. Quant. Spectrosc. Radiat. Transf., 110, 533–572, 2009.

[20] Fleischmann, O. C., Hartmann, M., Burrows, J. P., and Orphal, J.: New ultraviolet absorption cross-sections of BrO at atmospheric temperatures measured by time-windowing Fourier transform spectroscopy, J. Photoch. Photobio. A, 168, 117–132, 2004.

[21] Ryu, Y-H., Hodzic, A., Descombes, G., Hall, S., Minnis, P., Spangenberg, D., Ullmann, K., and Madronich, S.: Improved modeling of cloudy-sky actinic flux using satellite cloud retrievals, Geophys. Res. Lett., 44, 1592-1600, doi:10.1002/2016GL071892, 2017.

[22] Badosa, J., Gonzalez, J-A., and Calbo, J.: Using a parameterization of a radiative transfer model to build high-resolution maps of typical clear-sky UV index in Catalonia, Spain, Journal of Applied Meteorology, 789-803, 2005.

[23] Koepke, P., Bais, A., Balis, D., Buchwitz, M., De Backer, H., de Cabo, X., Eckert, P., Eriksen, P., Gillotay, D., Heikkila, A., Koakela, T., Lepeta, B., Litynska, Z., Lorente, J., Mayer, B., Renaud, A., Ruggaber, A., Schauberger, G., Seckmeyer, G., Seifert, P., Schmalwieser, A., Schwander, H., Vanicek, K., and Weber, M.: Comparison of models used for UV index calculations, Photochemistry and photobiology, 67, 657-662, 1998.

[24] Palancar, G. G., Lefer, B. L., Hall, S. R., Shaw, W. J., Corr, C. A., Herndon, S. C., Slusser, J. R., and Madronich, S.: Effect of aerosols and NO2 concentration on ultraviolet actinic flux near Mexico city during MILAGRO: measurements and model calculations, Atmos. Chem. Phys., 13, 1011-1022, doi:10.5194/acp-13-1011-2013, 2013.

[25] Xia, X., Zong, X., Cong, Z., Chen, H., Kang, S., and Wang, P.: Baseline continental aerosol over the central Tibetan plateau and a case study of aerosol transport from South Asia, Atmos. Environ., 45, 7370-7378, doi: 10.1016/j.atmosenv.2011.07.067, 2011.

[26] Xing, C., Liu, C., Wang, S., Hu, Q., Liu, H., Tan, W., Zhang, W., Li, B., and Liu, J.: A new method to determine the aerosol optical properties from multiple-wavelength $O_4$ absorptions by MAX-DOAS observation, Atmos. Meas. Tech., 12, 3289-3302, doi:10.5194/amt-12-3289-2019, 2019.